# The first juvenile dromaeosaurid (Dinosauria: Theropoda) from Arctic Alaska

Alfio Alessandro Chiarenza[1,2]*, Anthony R. Fiorillo[3], Ronald S. Tykoski[4], Paul J. McCarthy[5], Peter P. Flaig[6], Dori L. Contreras[4]

1 Imperial College London, London, United Kingdom, 2 University College London, London, United Kingdom, 3 Huffington Department of Earth Sciences, Southern Methodist University, Dallas, TX, United States of America, 4 Perot Museum of Nature and Science, Dallas, TX, United States of America, 5 Department of Geosciences, University of Alaska, Fairbanks, AK, United States of America, 6 Jackson School of Geosciences, Bureau of Economic Geology, University of Texas at Austin, Austin, TX, United States of America

* a.chiarenza15@gmail.com

## Abstract

Compared to the osteological record of herbivorous dinosaurs from the Late Cretaceous Prince Creek Formation of northern Alaska, there are relatively fewer remains of theropods. The theropod record from this unit is mostly comprised of isolated teeth, and the only non-dental remains known can be attributed to the troodontid cf. *Troodon* and the tyrannosaurid *Nanuqsaurus*. Thus far, the presence of members of Dromaeosauridae has been limited to isolated teeth. Here we describe a symphyseal portion of a small dentary with two ziphodont teeth. Based on tooth shape, denticle morphology, and the position of the Meckelian groove, we attribute this partial dentary to a saurornitholestine dromaeosaurid. The fibrous bone surface, small size, and higher number of mesial denticles compared to distal ones point to a juvenile growth stage for this individual. Multivariate comparison of theropod teeth morphospace by means of principal component analysis reveals an overlap between this dentary and Saurornitholestinae dromaeosaurid morphospace, a result supported by phylogenetic analyses. This is the first confirmed non-dental fossil specimen from a member of Dromaeosauridae in the Arctic, expanding on the role of Beringia as a dispersal route for this clade between Asia and North America. Furthermore, the juvenile nature of this individual adds to a growing body of data that suggests Cretaceous Arctic dinosaurs of Alaska did not undergo long-distance migration, but rather they were year-round residents of these paleopolar latitudes.

## Introduction

Dromaeosauridae [1–3] (S1 Table) is a group of predatory theropod dinosaurs evolutionarily close to the origin of birds [4, 5]. This clade likely originated in the Middle Jurassic [6, 7], with the first definitive dromaeosaurids recovered from Cretaceous deposits [8]. By the Late Cretaceous they reached a virtually cosmopolitan distribution [9], so far remaining unknown only in Antarctica. Given the small to medium size of most dromaeosaurids, and their fragile,

**Data Availability Statement:** All relevant data are within the manuscript and its Supporting Information files.

**Funding:** ARF received funding for this project from the National Science Foundation (OPP 0424594 to Fiorillo) and the National Geographic Society (W221-12 to Fiorillo). ARF received additional funding through the Perot PaleoClub, a private donation. The Perot Paleo Club played no role in the study design, data collection and analysis, decision to publish, or preparation of the manuscript.

**Competing interests:** The authors have declared that no competing interests exist.

highly pneumatic skeletons that are subject to greater incompleteness bias than many other dinosaur taxa [10, 11], complete remains of this group are generally rare and confined to exceptionally productive fossil localities [e.g. 12, 13]. North American taxa belong to at least 4 recognized major subclades (Dromaeosaurinae, Microraptorinae, Saurornitholestinae, and Velociraptorinae; S1 Table [14]) with probable Asian origins based on phylogenetic inference and local abundance of taxa referred to these clades [8]. Since the earliest discoveries of dinosaur remains on the Alaskan North Slope [15–17], the number of studies describing dinosaurs from the Prince Creek Formation and their role in clarifying paleobiogeographic and paleoecological aspects of the Cretaceous Arctic has greatly increased [e.g. 18–25].

Dinosaur teeth often preserve more easily and are more frequently recovered than bones [26], and the discovery of isolated teeth referable to Dromaeosauridae in many Late Cretaceous microsites has often provided important biogeographic data confirming the presence of the group in areas for which purely osteological remains are unknown [9, 27]. For example, Fiorillo and Gangloff [28] reported on isolated dromaeosaurid teeth from the Prince Creek Formation of Alaska, tentatively referring them to *Dromaeosaurus* and *Saurornitholestes*. Given the intermediate paleogeographic position of Alaska (as part of the ancient Beringian landmass), and its role as a land bridge between Asia and North America, additional dromaeosaurid remains with better resolved taxonomic identification have the potential to increase our understanding of the origin and dispersal of these clades through Asiamerica. Here we describe the first non-dental, osteological material of a saurornitholestine dromaeosaurid from Alaska, representing a unique morphotype. This find supports the presence of this clade in the Upper Cretaceous (lower Maastrichtian) Prince Creek Formation on the North Slope of Alaska (70˚ N, Fig 1).

## Materials and methods

DMNH 21183 is a symphyseal portion of a theropod dentary with a semi-erupted tooth and a replacement tooth preserved. The specimen was studied using a Nikon SMZ motorized stereomicroscope (Nikon Instruments, Inc., Melville, NY, USA), equipped with epi-fluorescence with an X-Cite XYLIS light source (Excelitas Technologies, Waltham, MA, USA) and GFP filter. Imaging of the specimen, including focal stacking and 3-D reconstructions, was completed using an attached Nikon Ri2 color CMOS camera with Nikon's NIS-Elements Acquisition and Analysis Software. Additional microscope observations and imaging were carried out using a Keyence Digital VHX-7000 series microscope (Keyence Corporation of America, Itasca, IL, USA). Dental nomenclature and terminology is based on Hendrickx et al. [29]. Terminology regarding ontogenetic characters is mostly, but not exclusively based on Sampson et al. [30], Carr [31], and Hone et al. [32]. Anatomical description is based on morphological observation by three of the authors (AAC, AF, RT). Comparisons were made based on first hand observations of relevant material by AAC, AF, and RT, as well as literature comparisons. Stratigraphic and sedimentological observations were carried out by three of the authors (AF, PF, PM) between 2005 and 2014 (Fig 2). Cladistic analyses and character evaluation were conducted by authors AAC, DC, RT, and AF.

To assess the systematic position of DMNH 21183 within Theropoda, we performed three different phylogenetic analyses. The first analysis, based on osteological and dental characters, used the dataset from Lee et al. [33], which included 120 operational taxonomic units (OTUs) and 1529 characters. See Lee et al. [33] and Cau et al. [34] for further information on character choice and coding. Our updated matrix differed only in the addition of DMNH 21183 as an OTU. The second and third analyses followed the protocols outlined by Hendrickx et al. [35] to identify isolated theropod teeth. DMNH 21183 was scored in one matrix based on dentition

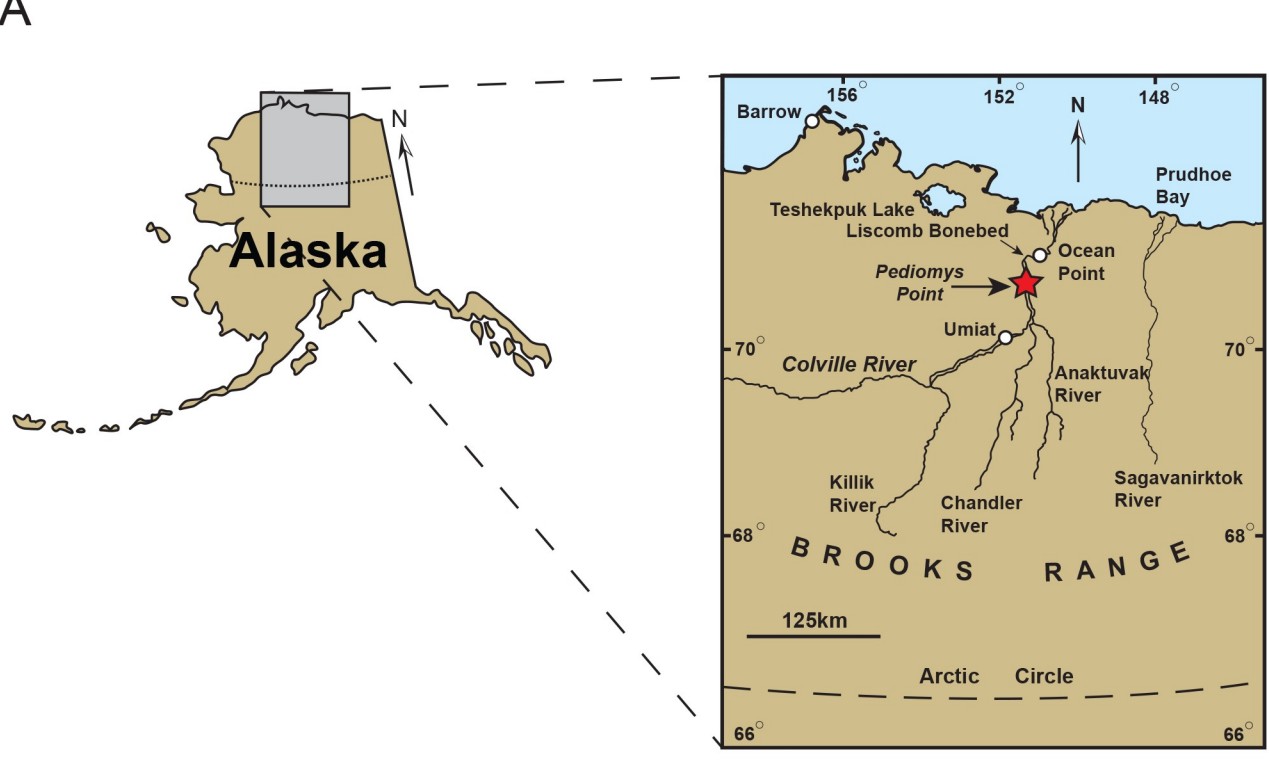

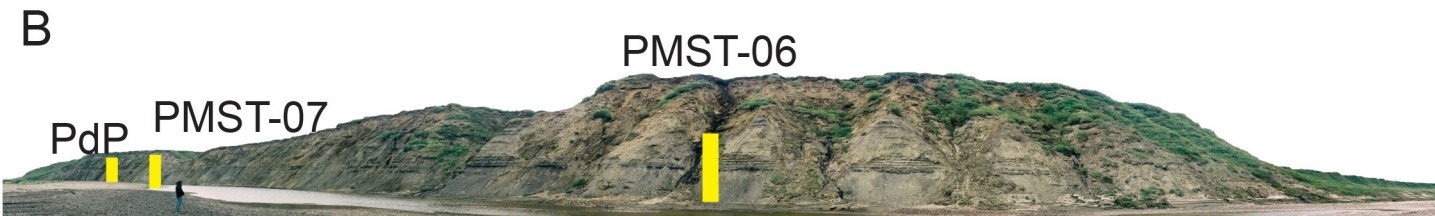

**Fig 1.** Locality map (A) of Pediomys point (red star) in the North Slope of Alaska, USA. Coordinates: N 70.018667˚, W 151.591488˚ (Paleocoordinates from paleobiodb.org: N 89.13˚, W -104.73˚). Stratigraphic sections schematized in Fig 2 are reported here in B, with PdP representing the fossil bone-bearing section.

characters, and another on tooth-crown-only characters. Details on character choice and rationale can be found in Hendrickx et al. [35], where original datasets are also reported as supplementary material. The dentition-based matrix consists of 107 taxa and 146 characters. The tooth-crown-based matrix includes 102 taxa and 91 characters. These dental datasets were analyzed by additionally forcing a topological constraint following Hendrickx et al. [35], reflecting a previously recovered tree topology for Theropoda (e.g. Rauhut and Carrano [36] and Brusatte et al. [37]), with DMNH 21183 as a floating OTU. Character scorings for DMNH 21183 in all three phylogenetic datasets are reported in Supplementary Information Document (S1 Dataset). Characters were all equally-weighted and treated as unordered or ordered following the source literature. The phylogenetic analyses were run using the software TNT [38]. For each dataset, we performed a "New Technology" search that included a combination of random and constraint sectorial searches, ratchet, tree-drifting, and tree-fusing, with ten search replications as the starting point for each hit and searches carried out until 100 hits of the same minimum tree length were achieved (TNT command used is "xmult = hits 100 replic 10 css rss

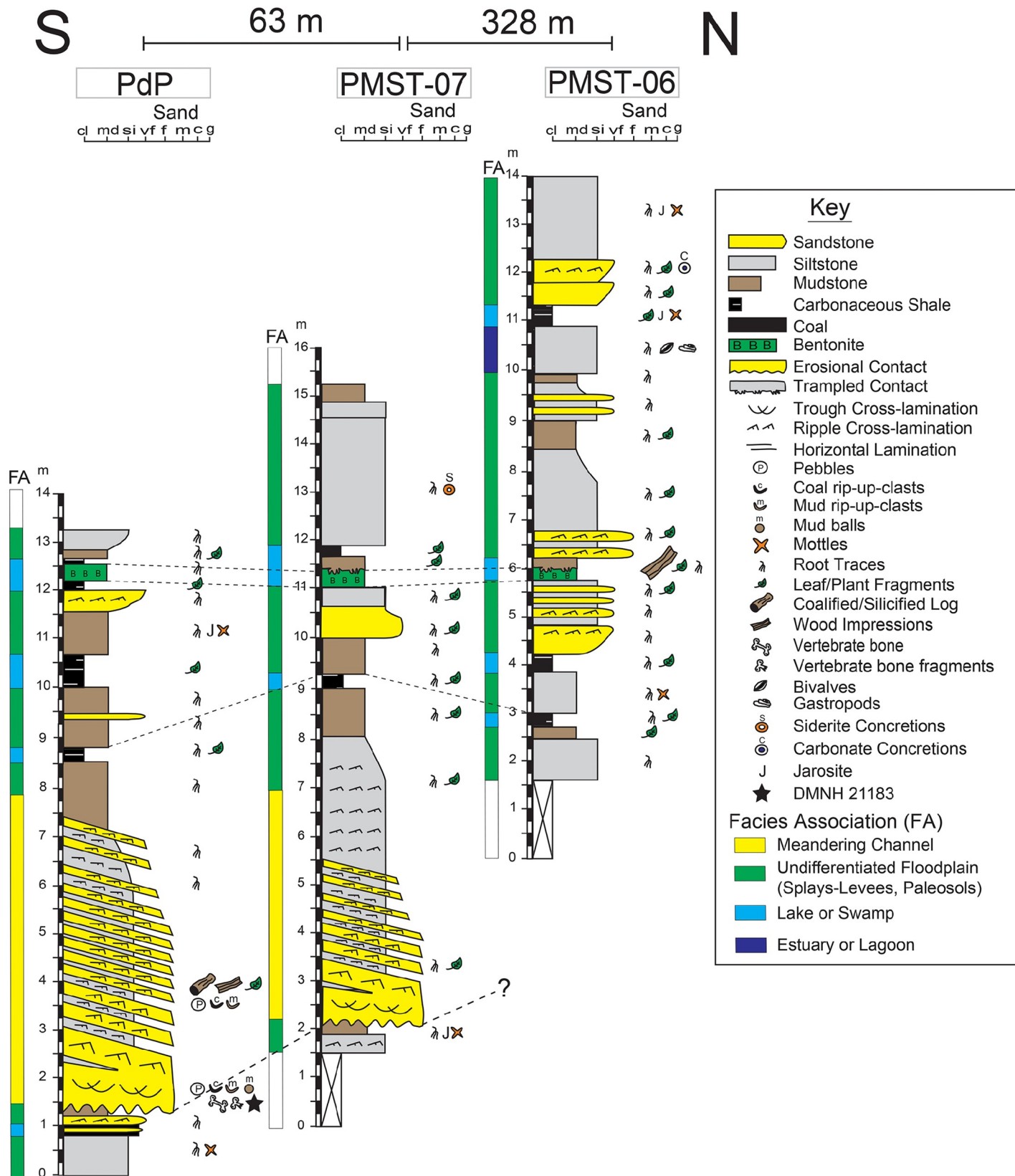

**Fig 2. Geological setting.** Stratigraphic sections at Pediomys point. Black star represents DMNH 21183.

ratchet 5 drift 5 fuse 5"). The most parsimonious trees (MPTs) obtained were subjected to two rounds of TBR branch swapping (command "bbreak = TBR"). Strict consensus trees were generated from the resulting set of MPTs Nodal support was calculated for the consensus by running a standard bootstrap analysis with 1000 replicates.

To test the qualitative observations reported in the anatomical description section, we performed multivariate analyses of theropod tooth measurements including those retrieved by the erupted tooth in DMNH 21183 (S2 Dataset). Although this tooth is not fully erupted, the breaks on the medial side of the dentary around the area of the relative alveolus and observations under microscopy allowed measurement of the crown height of the tooth. Measurements on tooth morphometrics are from Gerke and Wings [39] and a modified dataset from Larson and Currie [40] (S2 Dataset). The dataset from Gerke and Wings [39] includes measurement data for 335 theropod teeth for which entries were modified to reflect a higher systematic rank or clade compared to the original genus-level classification (e.g. Abelisauridae, Dromaeosauridae, Tyrannosauridae). Measurements for this dataset include crown basal length (CBL), crown height (CH), crown basal width (CBW), and the ratio between mesial (anterior) and distal (posterior) denticles (DSDI).

The dataset from Larson and Currie [40] includes measurement data for over 1200 small theropod teeth, mainly from the latest Cretaceous of the Western Interior Basin of North America, and has also been used for similar studies in the past [e.g. 41, 42]. Principal measurements included fore-aft basal length (FABL), crown height (CH), basal width (BW), and anterior (ADM) and posterior denticles per millimeter (PDM). This dataset was modified by removing all entries with at least one variable missing or unable to be evaluated, such as the lack of a measurement caused by the absence of the related structure (e.g. ADM in teeth without mesial denticles or carina). In this dataset, the taxonomy of the identified teeth was referred to the "family-ranked" clade with the exception of *Richardoestesia*. For example, *Saurornitholestes langstoni* and *Atrociraptor marshalli* are both referred to Saurornitholestinae, following [14]. The Milk River cf. *Zapsalis* is referred to *Saurornitholestes* following Currie and Evans [43]. The Aquilan cf. *Richardoestesia gilmorei*, the Oldman cf. *Richardoestesia gilmorei*, the Horseshoe Canyon cf. *Richardoestesia*, the Lancian cf. *Richardoestesia*, the Aquilan cf. *Richardoestesia isosceles*, and *Richardoestesia isosceles* are all referred to *Richardoestesia*. The subclades of theropods included in this dataset, apart from the specimen studied firsthand (DMNH 21183) are: Dromaeosaurinae, *Richardoestesia*, Saurornitholestinae and Troodontidae.

We performed two principal components analyses (PCA), a multivariate technique that takes a number of measurements and converts them into a smaller set of values that represent the variability of the sample plotted in a multivariate space. Following the example of similar studies (e.g. [9, 35, 39]), the same measurements were also used for a linear discriminant function analyses (DFA) on both the datasets. This method provides a value to assess the degree of confidence on the classification of the clusters in the morphospace, where 0.5 is no better than random in model accuracy while 1 represents perfect accuracy (100% accuracy [44]). Both these methods allow the dataset of teeth to be plotted in a morphospace, and to quantitatively compare the degree of overlap and the relative position of DMNH 21183 with 1) the main clades of Theropoda from the Gerke and Wings [39] dataset and 2) the main deinonychosaurian subclades (e.g., Dromaeosaurinae, Troodontidae) from the Larson and Currie [40] dataset. Multivariate analyses were performed in the software "R" with the "MASS" package [45]. PCA outcomes are reported as results in the relevant section while DFA plots are provided in Supplementary Information (S2 and S3 Figs). See S1 Table for systematic definitions of clades and lineages mentioned here and used throughout the text.

### Institutional abbreviations

AMNH–American Museum of Natural History, New York City, USA; DMNH–Perot Museum of Nature and Science, Dallas, Texas, USA; IVPP–Institute for Vertebrate Paleontology and Paleoanthropology, Beijing, China; MPC-D–Institute of Paleontology and Geology, Mongolian Academy of Sciences (formerly known as IGM), Ulaanbaatar, Mongolia; YPM–Yale Peabody Museum of Natural History, Yale, Connecticut, USA; NHMUK PV–Natural History Museum, London, UK; TMP–Royal Tyrrell Museum of Palaeontology, Drumheller, Alberta, Canada.

## Results

### Locality

The specimen was collected from exposures of the Prince Creek Formation, at the Pediomys Point locality (Figs 1 and 2). The locality is along the Colville River, 8 km upstream from the Liscomb bonebed, North Slope Borough, Alaska, USA (Fig 1). Bulk sediment was collected at the site over multiple field seasons between 2005–2007, 2012, and 2014, with screenwashing and sorting of the material conducted at the Perot Museum of Nature and Science (DMNH) in Dallas, Texas, USA.

### Geological setting and depositional environments

The Prince Creek Formation (PCF) was deposited in the Colville Basin of northern Alaska and provides us with the largest collection of polar dinosaur bones in the world [25]. The PCF was originally sub-divided into two subunits or tongues: an older Tuluvak Tongue and a younger Kogosukruk Tongue [46]. However, Mull et al. [47] revised this nomenclature based on regional stratigraphic correlations and the Prince Creek Formation was redefined to include only the former Kogosukruk Tongue along with some younger, Paleocene strata. The total thickness of the PCF along the Colville River is approximately 450 m [48, 49]. Biostratigraphic [50–59], and isotopic analyses [60] indicate that the age of the Prince Creek Formation ranges from Campanian to Paleocene. All deposits containing evidence of dinosaurs are Early Maastrichtian in age and approximately 68.5–70 million years old [57, 59, 61–63].

The PCF is an alluvial succession composed of conglomerate, sandstone, siltstone, mudstone, carbonaceous shale, coal, and bentonite [59, 61, 64]. The mostly fine-grained sediments record deposition in low energy, suspended load channels and on organic-rich floodplains on a low-gradient coastal plain. Thicker, multi-story meandering trunk channels contain the largest grain sizes and record the highest flow velocities in the area. Smaller meandering and fixed or anastomosed distributary channels formed crevasse splay-complexes adjacent to trunk channels. Abundant organic-rich facies were deposited in low-lying areas between the large channels and splay-complexes. Floodplains contained levees, splays, lakes, ponds, swamps, and soil-forming environments. Volcanic ashfall was common and smectite-rich bentonites were commonly preserved in wet floodplain environments [65]. Trampling of sediments by dinosaurs was common [63]. Weakly-developed paleosols similar to modern *Entisols*, *Inceptisols*, andic soils, and potential acid sulfate soils formed on levees, point bars, crevasse splays, and along the margins of floodplain lakes, ponds, and swamps that also supported lowland trees, shrubs, herbs, ferns, moss, and algae [61, 65]. Macroscopic and micromorphological features, and illite-smectite mixed-layer clay minerals in paleosols indicate predominantly waterlogged, reducing conditions interrupted by oxidizing conditions and periodic drying out of some soils [61, 62, 64, 65]. Soils experienced repeated sediment influx from overbank flooding of nearby distributary channels, and periodic deposition of hyperconcentrated flows [62].

Sediments deposited in the most distal areas of the coastal plain contain evidence of marine influence that includes inclined heterolithic stratification in channel point bars and pyrite, jarosite mottles, jarosite halos surrounding root-traces, and brackish-water fauna in floodplain facies [64, 66].

The 750 m-long outcrop belt at Pediomys Point is located near the upriver end of a slough off the main course of the Colville River (Fig 1). Cretaceous environments preserved at Pediomys Point include a meandering distributary channel that transitions laterally into a silt and mud-filled abandoned channel along with floodplain environments that include crevasse splays, levees, small lakes and swamps, floodplain paleosols, and ashfall deposits ([67], Fig 2). Trampled sediments found above a bentonite (Fig 2) are similar to those described from strata 4.25 kilometers downstream along the Colville River that are interpreted as adult and juvenile hadrosaur tracks along a swamp margin [63]. Rare brackish-water clams (most likely *Nucula* aff. *N. percrassa* Conrad; see [66]) and gastropods found near the top of the stratigraphic succession at Pediomys Point suggest an estuarine or lagoonal environment for those deposits [66]. Interfingering of continental-terrestrial and shallow marine deposits, including those of flood basins, interdistributary bays and estuaries were identified in older deposits of the PCF along the Colville River [68] and in younger deposits above the Liscomb Bonebed near Ocean Point [69]. This suggests that the environments at Pediomys Point were transitional between the subaerial coastal plain or delta plain, and shallow marine habitats (Fig 2).

## Systematic paleontology

Dinosauria [70]
Theropoda [71]
Dromaeosauridae [1]
Eudromaeosauria [14]
Saurornitholestinae [14] indet.

## Referred specimen

DMNH 21183. The anterior portion of a right dentary, preserving two teeth and four alveoli (Fig 3).

## Description

DMNH 21183 (Fig 3) (Table 1) is an anterior portion of a right dentary with an unerupted mesial tooth (rdt2; Fig 3A–3C) and a distally placed partially erupted tooth (rdt3; Fig 3). The anterior surface of the dentary is damaged, obscuring details of the symphyseal region. Given the position of the erupted tooth (rdt3; Fig 3) above the Meckelian foramina (on the medial surface of the dentary, Fig 3B), and above an the anteroventral process of the dentary (ave; Fig 3A and 3B), we identify this as the 3rd dentary tooth in the dentary [e.g. 43, 72], with the more mesially positioned, unerupted tooth (rdt2) being identified as the 2nd. Alveoli 2–4 are preserved (Fig 3A–3C), although the margins of the latter (fourth) are obscured by erosion. Both the medial and lateral surfaces of the dentary present a parallel, anteroposteriorly-oriented, fibrous bone texture. The anterior margin of the 3rd alveolus (a3; Fig 3A–3C) has a raised anterior rim that extends dorsally up to approximately mid-height of the crown (a3). There is a well-preserved subtriangular interdental plate lingual to the septum separating the 2nd and 3rd alveoli (a2–a3), and a bigger, semicircular one between the 3rd and 4th alveoli (a3–a4). The paradental space is relatively dorsoventrally short. Laterally, the dorsal rim of the dentary rises into a triangular ridge, which creates a convex surface anterolaterally that sinks at its base into a circular fossa. The alveoli are elliptically shaped, being slightly anteroposteriorly longer than

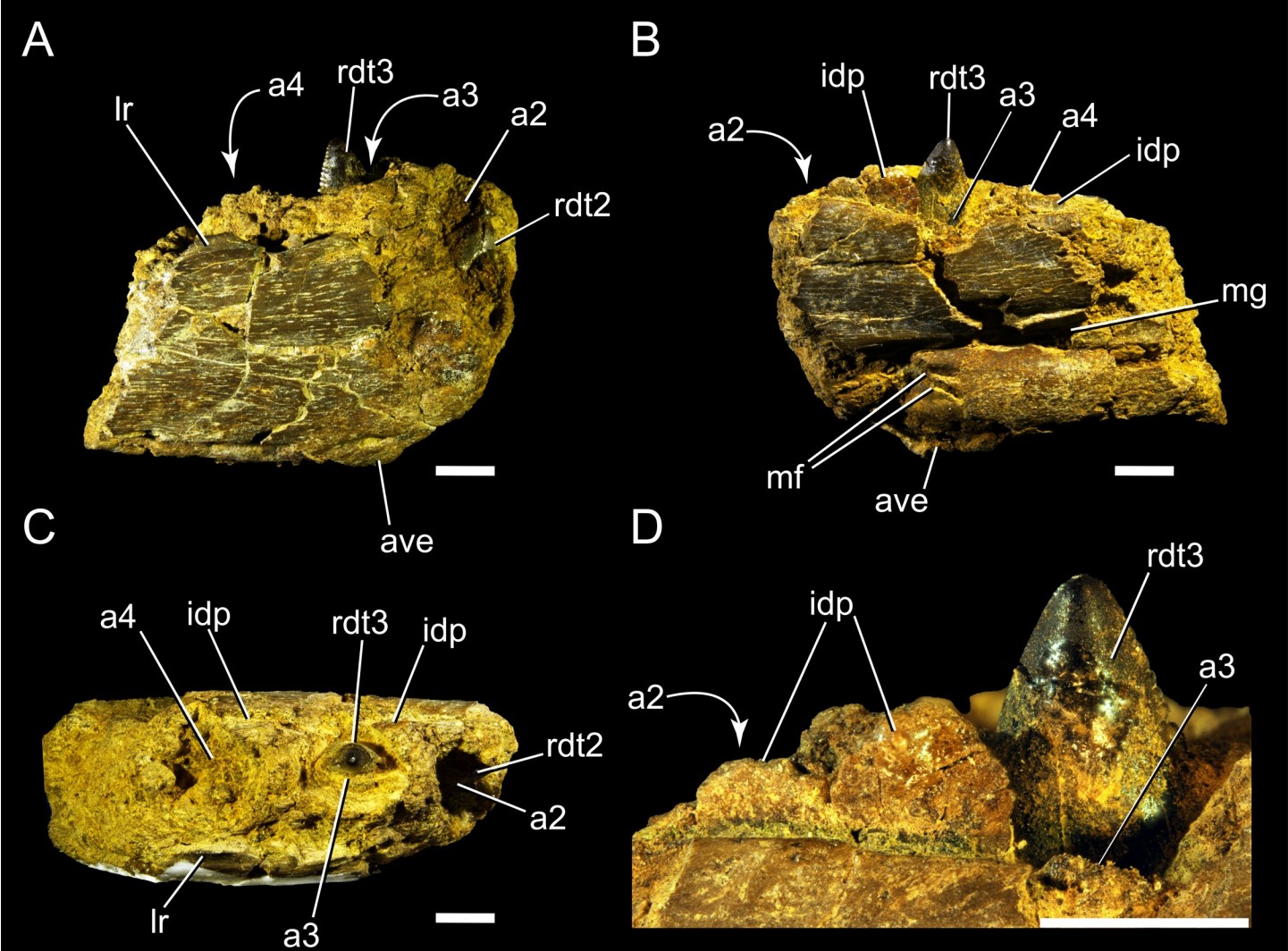

**Fig 3. DMNH 21183.** Anterior portion of a dromaeosaurid dentary in lateral (A), medial (B), dorsal view (C) views and close up of the interdental plates and 3rd tooth in medial view (D). Curved arrows represent features hidden on that view. Abbreviations: a2, 2nd alveolus; a3, 3rd alveolus; rdt2, 2nd dentary tooth; rdt3, 3rd dentary tooth; rdt4, 4th dentary tooth; ave, antero-ventral process; idp, interdental plate; lr, lateral ridge; mg, Meckelian groove; mf, Meckelian foramina. Scale bar: 2 mm.

mediolaterally wide (Fig 3C). The medial side of the dentary features a shallow Meckelian groove (Fig 3B), set slightly more ventrally than mid-height of the bone. The medial side of the dentary features an anteriorly located Meckelian foramen (Fig 3B), which may be paired with a second, more ventrally positioned foramen, but damage to the bone surface in this area makes the identification of this feature uncertain. There is an anteroventral expansion in the alveolar margin of the dentary, visible both medially and laterally (Fig 3A and 3B) that is excavated by a sub-oval fossa on the lateral side (Fig 3A).

Both preserved teeth in DMNH 21183 are ziphodont. The mesiodistal axis of the 2nd tooth (rdt2) is more anteromedially oriented in relation to the lateral margin of the dentary. The 2nd tooth is unerupted, but damage to the medial surface of the alveolar wall exposes the most apical half of the tooth crown. While the anterior alveolar margin covers the mesial surface of the 2nd tooth (rdt2), 13 denticles are visible along the distal carina (Fig 4A and 4B). The apical-

**Table 1. Measurements of DMNH 21183.**

| Elements measured | Measurements (mm) |
|---|---|
| Dentary anteroposterior length (dorsal view) | 14.34 |
| Dentary anteroposterior length (lateral view) | 14.59 |
| Maximum dentary mediolateral width | 6.15 |
| Maximum dentary dorsoventral depth | 9.44 |
| Alveolus II mesiodistal length | 2.95 |
| Alveolus II labiolingual width | 1.27 |
| Alveolus III labiolingual width | 2.76 |
| Alveolus III mesiodistal length | 3.71 |
| Tooth III (rdt3) mesiodistal width | 1.97 |
| Tooth III (rdt3) labiolingual length | 1.05 |
| Tooth III (rdt3) apicobasal length | 4.50 |

most half of portion of the 3$^{rd}$ tooth (rdt3) is erupted, with 12 denticles visible on the distal carina (Fig 3A, 3B and 3D).

The teeth have larger distal denticles than the mesial ones. The unerupted portion of the third tooth crown can be seen through a fracture on the medial side of the dentary (Fig 3B). Taking into account the base of this crown, an estimate of ~30 denticles per serrated carina can be inferred for the tooth.

Unfortunately, wear of the mesial carina destroyed some details of the denticles. However, apart from some denticles, most of the interdenticular grooves between adjacent denticles are clear, and they are shallow incisions rather than deep sulci (Fig 3D, S1 Fig). In the 2$^{nd}$ tooth this comparison between mesial and distal carina is not possible because the mesial margin of the tooth is hidden by sediment and the labial wall of the alveolus (Fig 4A and 4B). The shape of the apices of the distal denticles is slightly hooked, with an orientation toward the apex of the crown and with an externally rounded margin rather than with a sharp tip (Fig 4A–4C).

## Phylogenetic analysis results

Analysis of the Lee et al. [33] data matrix with DMNH 21183 added resulted in 384 MPTs that are 6043 steps long, with a Consistency Index (CI) = 0.244 and a Retention Index (RI) = 0.587. The strict consensus (Fig 5; S4 Fig) reproduced the topology hypothesized by Lee et al. [33]. DMNH 21183 is recovered in Dromaeosauridae excluding Unenlaginae, in a polytomy with most other eudromaeosaurs (*Dromaeosaurus* + *Utahraptor*) and Microraptorinae clades.

The addition of DMNH 21183 to the dental-only character matrix of Hendrickx et al. [35] produced 2 MPTs (1314 steps long, CI = 0.194, RI = 0.418). The Strict Consensus of the two trees recovers Paraves as a trichotomy between Avialae, Troodontidae and Dromaeosauridae (Fig 6A; S5 Fig). DMNH 21183 is recovered as the sister OTU of *Saurornitholestes*, in a partially resolved Eudromaeosauria (*sensu* Longrich and Currie [14]). This dromaeosaurid clade is represented by a polytomy between *Atrociraptor*, a monophyletic clade with *Deinonychus*, *Tsaagan* and *Velociraptor*, and another monophyletic clade with *Dromaeosaurus* and *Bambiraptor* as successively closer taxa to the clade *Saurornitholestes* + DMNH 21183.

Lastly, the analysis that includes DMNH 21183 in the tooth-crown-based data matrix of Hendrickx et al. [35] recovers 5 MPTs (867 steps, CI = 0.183; RI = 0.439). The strict consensus tree (Fig 6B; S6 Fig) produces a polytomy between alvarezsaurs, therizinosaurs, oviraptorosaurs and the rest of Maniraptora (Avialae, Troodontidae and Dromaeosauridae). DMNH 21183 is recovered within Eudromaeosauria (*sensu* Longrich and Currie [14]), in a polytomy

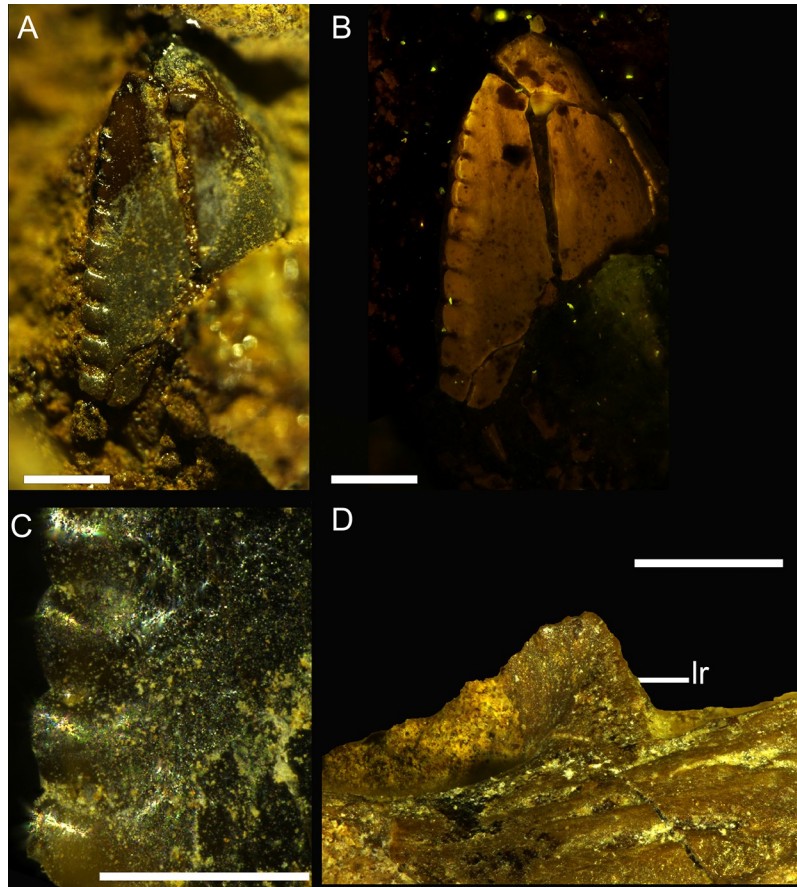

**Fig 4. Closeup on the dentary teeth and lateral ridge in DMNH 21183.** Detail of the 2nd dentary tooth highlighting the distal carina under normal light (A) and fluorescent microscopy (B). Details of the distal denticles (C) and close-up of the lateral ridge (lr) close to the alveolar margin in lateral views (D). Abbreviation: lr, lateral ridge. Scale bar: 0.5 mm.

between *Atrociraptor*, *Bambiraptor*, *Deinonychus*, *Saurornitholestes*, and an Asian Velociraptorinae clade (*Velociraptor + Tsaagan*).

## Multivariate analysis results

The dentition of DMNH 21183 (the 3rd and more exposed-better preserved tooth; rdt3 in Fig 3D) was assessed in a morphometric dataset of theropod teeth [39], with a PCA analysis returning four axes with the following eigenvalues and percentages of total variance explained by each axis: Axis 1 (0.377, 96.583%), Axis 2 (0.006, 1.594%), Axis 3 (0.004, 1.13%), Axis 4 (0.003, 0.69%). Coefficients for the five measurements on each axis are given in Table 2. The majority of the variance is captured in the first two axes of the principal components. The highest variable contribution is represented by CBW (37.79%), followed by CH (~33.70%), CBL (~28.12%), and lastly DSDI (0.38%). The position of DMNH 21183 in the first two axes of the theropod teeth morphospace is shown in Fig 7. DMNH 21183 overlaps the dromaeosaurid morphospace in the lower left quadrant of the plot. This convex hull partially overlaps with those of troodontids, noasaurids, and basal theropods among others. This cluster is set on the opposite side from the centroids of allosauroids, ceratosaurids, and spinosaurids, which occupy most of the center and right area of the teeth morphospace. DMNH 21183 is in the opposite area of the morphospace than tyrannosaurids (Fig 7). A similar spatial arrangement

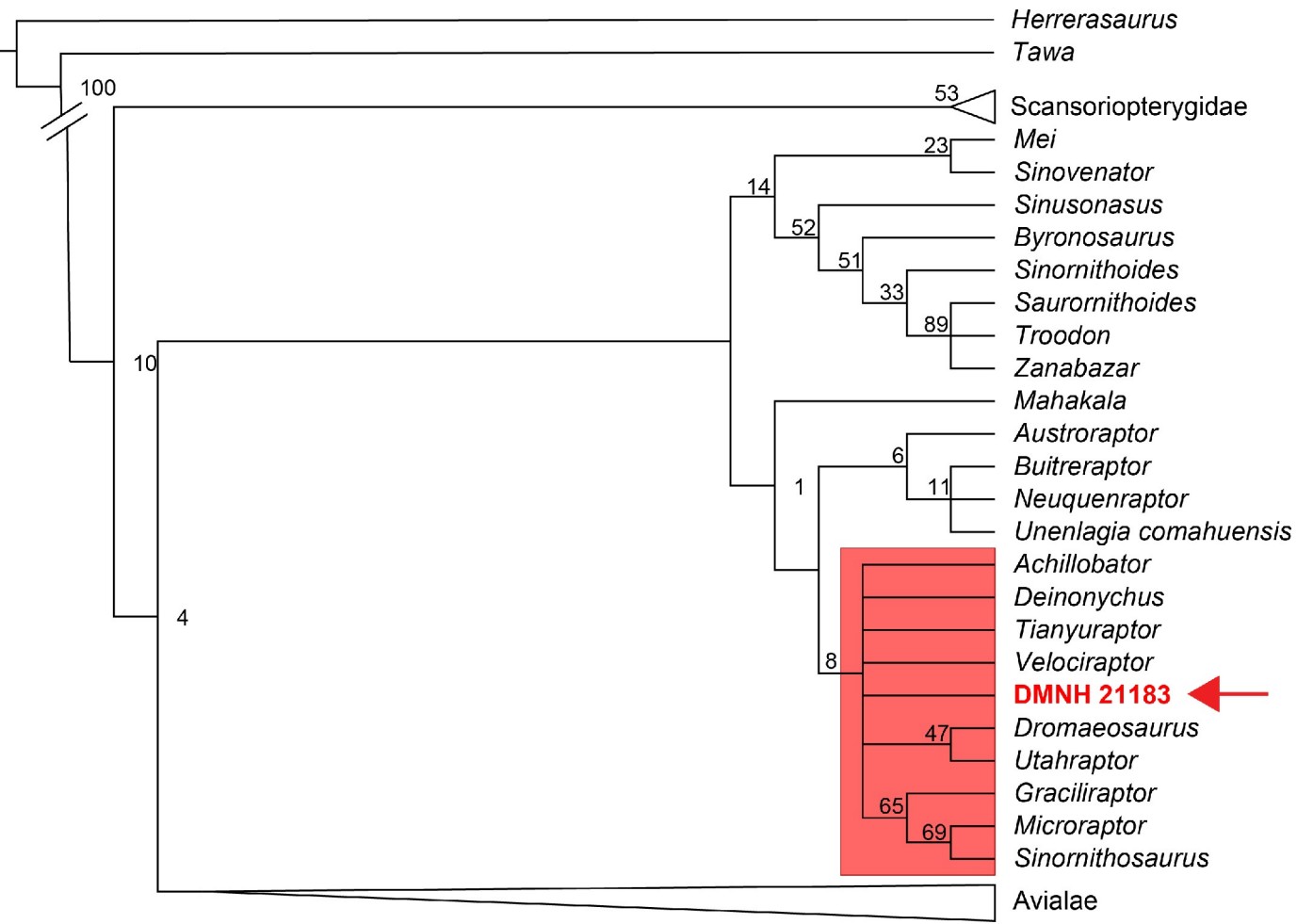

**Fig 5. Phylogenetic position of DMNH 21183.** Strict consensus topology of the shortest trees recovered by the parsimony analyses of the phylogenetic dataset of Lee et al. [33] with the addition of DMNH 21183 (384 MPTs, 6043 steps, CI = 0.244, RI = 0.587). The main clades of Theropoda outside Deinonychosauria are collapsed for space constraints. Full topology available in S4 Fig. Numbers adjacent to nodes are the bootstrap values. Red box highlights the node containing DMNH 21183 (red arrow).

is obtained with the DFA analysis (S2 Fig), in which model accuracy for classification of DMNH 21183 as a dromaeosaurid is of 0.65. To further explore the position of DMNH 21183 between Dromaeosauridae and the other taxa that were morphometrically close in this first set of multivariate analyses, we used a dataset [40] of deinonychosaurian teeth (see Materials and Methods) and performed a PCA analysis. The analysis returned four axes with the following eigenvalues and percentages of total variance explained by each axis: Axis 1 (15.005, 77.175%), Axis 2 (3.584, 18.433%), Axis 3 (0.559, 2.879%), Axis 4 (0.294, 1.513%). Coefficients for the five measurements on each axis are given in Table 3.

Coefficients of the PCA analysis run on the theropod teeth dataset published by Gerke and Wings [39] with the addition of DMNH 21183. ADM, anterior denticles per millimeter; BW, basal width; CBL, crown base length (in mm); CBW: crown base width; CDA: crown distal angle; CH: crown height (in mm); CH, crown height; DSDI: denticle size difference index, denticles in mesial carina divided by those in distal carina.

Coefficients of the PCA analysis run on the deinonychosaurian teeth dataset published by Larson and Currie [40] with the addition of DMNH 21183. ADM, anterior denticles per

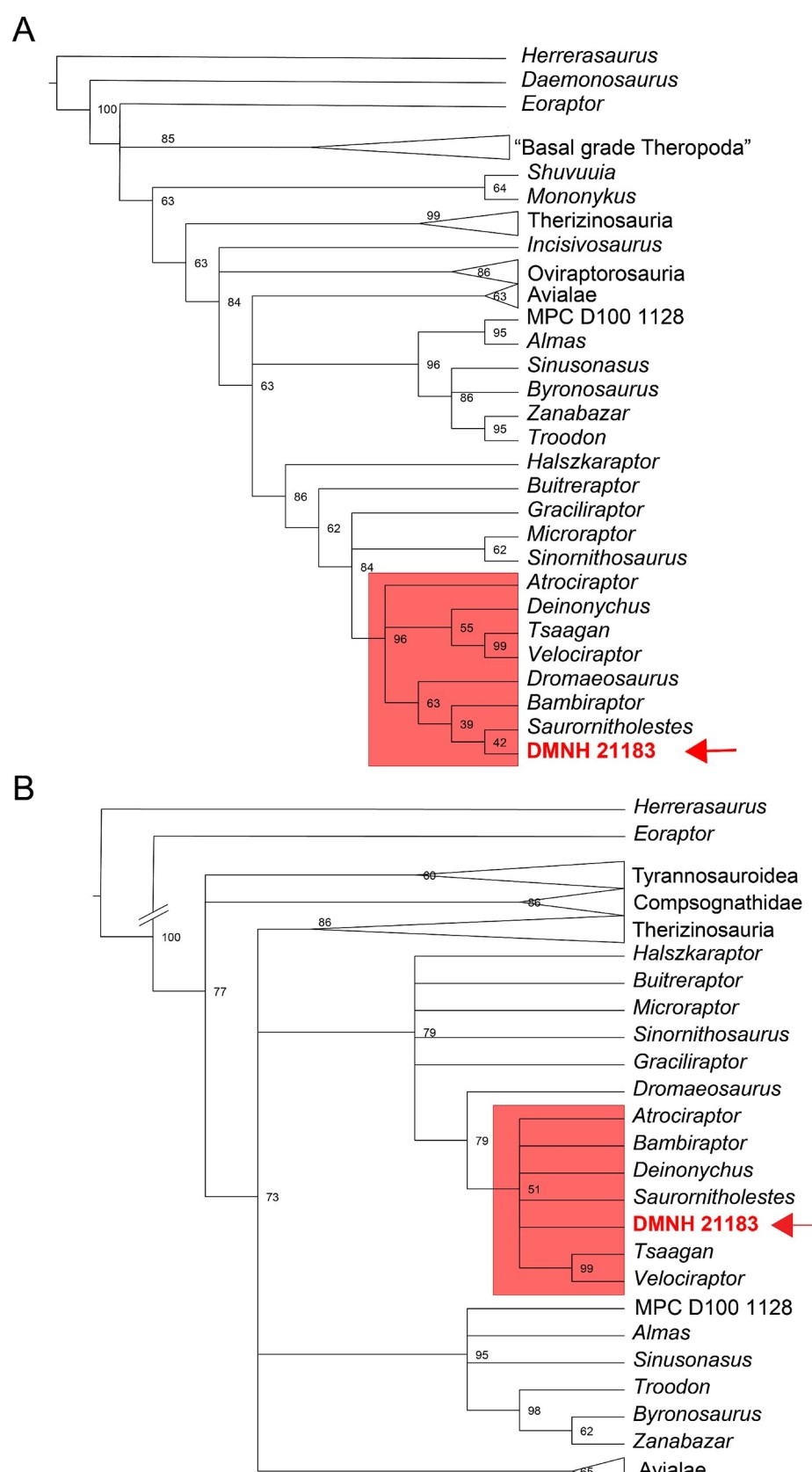

**Fig 6. Phylogenetic position of DMNH 21183 in Hendrickx et al. [35].** Strict consensus topology of the shortest trees recovered by the parsimony analyses showing the position of DMNH 21183 in (A) the dentition-only character matrix from Hendrickx et al. [35] (2 MPTs, 1314 steps, CI = 0.194, RI = 0.418), and (B) the tooth-crown-only character matrix from Hendrickx et al. [35] (5 MPTs, 867 steps, CI = 0.183, RI = 0.439). The overall topology was constrained in both analyses, with DMNH 21183 allowed to float. The main clades of Theropoda outside Deinonychosauria are collapsed for space constraints. Full topology available in S5 and S6 Figs. Numbers adjacent to nodes are the bootstrap values. Red box highlights the node containing DMNH 21183 (red arrow) in Eudromaeosauria.

millimeter; BW, basal width; CH, crown height; FABL, fore-aft basal length; and PDM, posterior denticles per millimeter.

The majority of the variance is explained by the first two axes of the principal components. The highest variable contribution (similarly to previous analyses on this dataset [40–42]) is represented by CH (~60%) followed by PDM (~30%), FABL (~15%) and lastly ADM (<5%). The position of DMNH 21183 in the first two axes of the deinonychosaurian teeth morphospace is shown in Fig 8. DMNH 21183 overlaps the Saurornitholestinae morphospace, being set further away from the centroid clustering the other analyzed taxon with hooked denticles, Troodontidae. It is outside the convex hull comprising Dromaeosaurinae, which has mostly subequal, rectangular denticles, and is set within the Saurornitholestinae morphospace in a position slightly toward the *Richardoestesia* morphotype. DFA results in a spatial arrangement of data points similar to the one of the PCA (S3 Fig), and the analyses provides a model accuracy of 0.8 for classification of DMNH 21183 as a saurornitholestine deinonychosaur. This outcome is comparable to those generated by the phylogenetic analyses, and we confidently refer DMNH 21183 to the Saurornitholestinae. We do not at this time assign it to any currently recognized species within the clade.

One major caveat of these analyses regards the potential bias due to the different ontogenetic stages of the teeth included in the sample [41], and most importantly in relation to the likely juvenile growth stage of DMNH 21183. Because PCA analysis would simply group observations based on measurements, it is likely that teeth belonging to juvenile and adult individuals will cluster in separate areas of the morphospace (see Discussion for a more in depth comparison of denticle size). This issue has been shown in the literature (e.g. [73, 74]) to affect previous PCAs of tyrannosauroid teeth, so we use this morphometric line of evidence as a complementary tool to assess the systematic identification of DMNH 21183.

## Discussion

### Morphological comparisons and the phylogenetic position of DMNH 21183

While DMNH 21183 is fragmentary, there are enough anatomical characters preserved in it to indicate its likely systematic position. Although very few non-dental theropod remains have been found in the Prince Creek Formation of Alaska, a general comparison with other contemporaneous theropod taxa, with a particular focus on those clades previously recognized in the formation (Dromaeosauridae, Tyrannosauridae, Troodontidae) is attempted here. The

**Table 2. Coefficients of the PCA analysis run on the theropod teeth dataset published by Gerke and Wings [39].**

| Measurement | PC1 | PC2 | PC3 | PC4 |
|---|---|---|---|---|
| CBL | 0.530305 | -0.207765 | 0.108537 | 0.814757 |
| CBW | 0.614766 | 0.7250023 | -0.251823 | -0.181711 |
| CH | 0.580509 | -0.605216 | 0.065003 | -0.540829 |
| DSDI | -0.062033 | -0.254791 | 0.959468 | 0.1032189 |

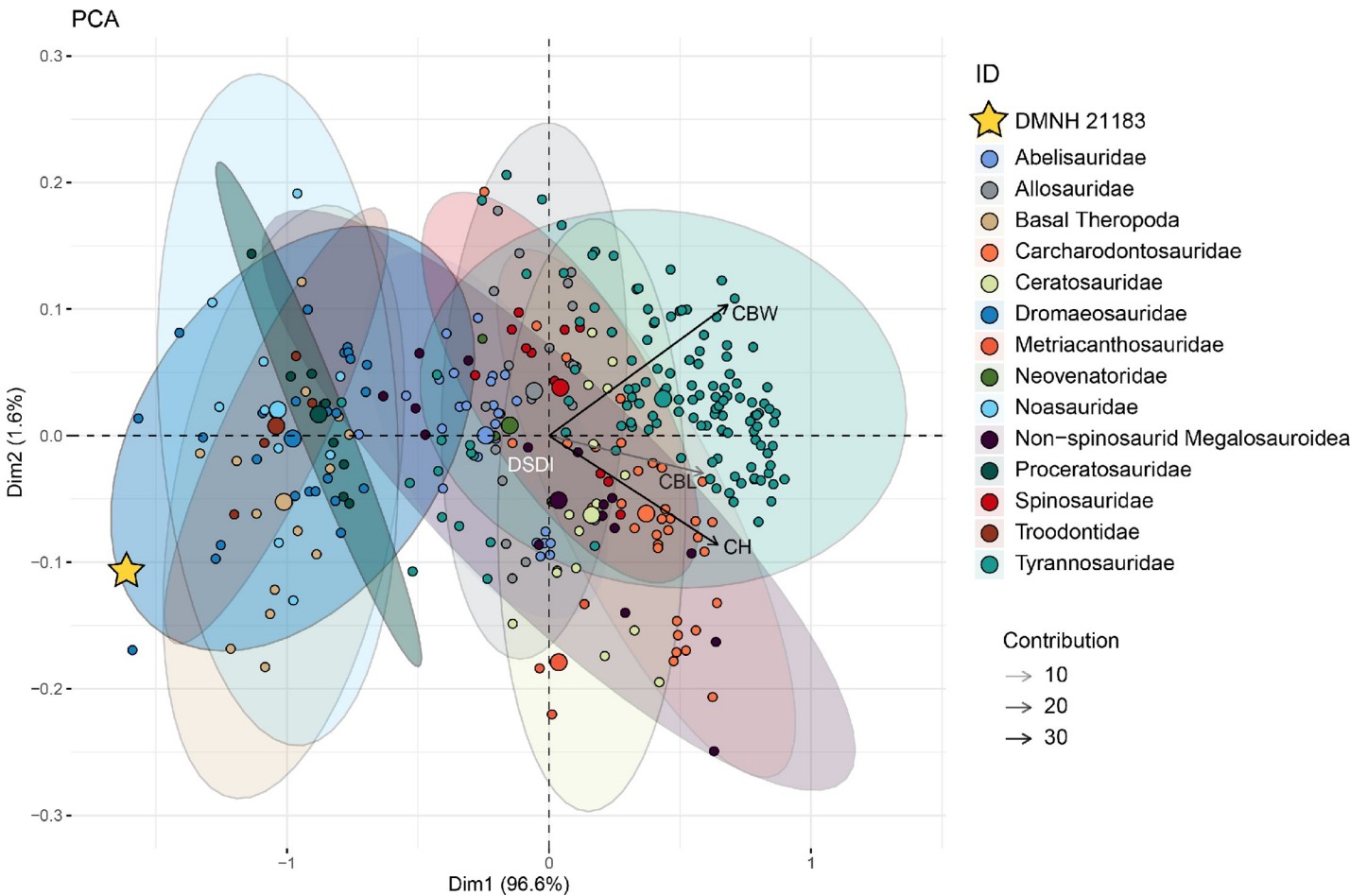

**Fig 7. Position of DMNH 21183 in the theropod teeth morphospace.** Principal components analysis of theropod teeth morphospace based on the dataset from Gerke and Wings [39]. DMNH 21183 is marked by a yellow star. Abbreviations: crown height (CH), crown basal width (CBW), ratio between mesial (anterior) and distal (posterior) denticles (DSDI). Morphometric dataset prepared as explained in methods and including measurements from DMNH 21183 reported in Supplementary Information (S2 Dataset).

combination of ziphodont dentition, presence of interdental plates (e.g. [75]), a Meckelian groove, and Meckelian foramina are characteristic of theropod dinosaurs (e.g. [76]). The orientation of the anterior-most tooth socket and its unerupted tooth (both anteromedially oriented), the presence of paired Meckelian foramina, and a ventral expansion laterally pierced by a fossa, point toward the identification of the specimen as an anterior portion (almost symphyseal, as also shown by the ventral enlargement of the dentary in lateral view) of a theropod dentary. The shallow Meckelian groove present in the specimen is a derived maniraptoran feature [77], in contrast to the deep groove seen in basal tyrannoraptorans (e.g.

**Table 3. Coefficients of the PCA analysis run on the deinonychosaurian teeth dataset published by Larson and Currie [30].**

| Measurement | PC1 | PC2 | PC3 | PC4 |
|---|---|---|---|---|
| FABL | 0.352856 | -0.03872 | 0.568433 | 0.742211 |
| CH | 0.838152 | 0.423491 | -0.31658 | -0.13392 |
| BW | 0.176441 | -0.00224 | 0.739112 | -0.65006 |
| PDM | -0.37665 | 0.90507 | 0.174281 | 0.092801 |

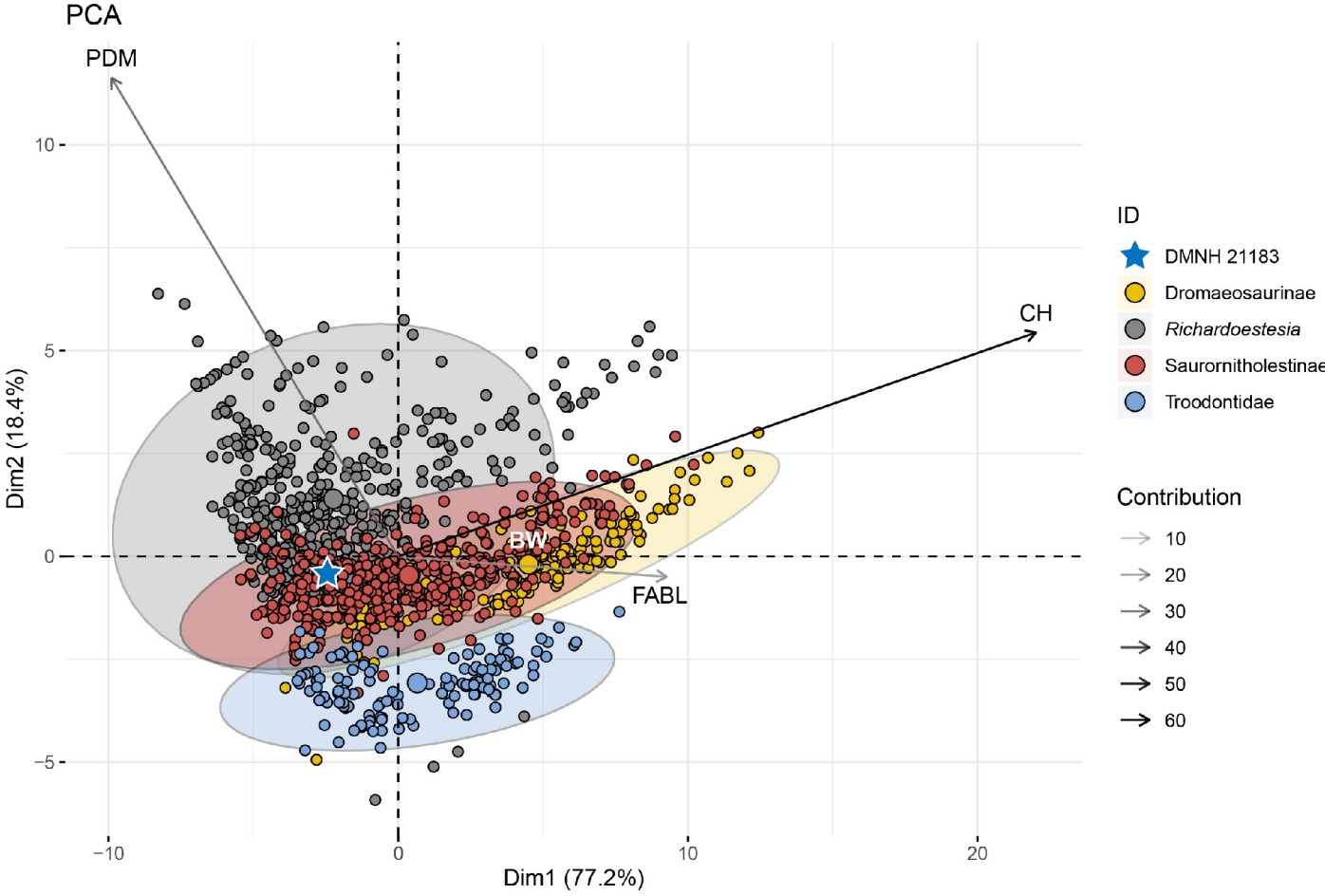

**Fig 8. Position of DMNH 21183 in the paravian teeth morphospace.** Principal components analysis of theropod teeth morphospace based on the dataset from Larson and Currie [30]. DMNH 21183 is marked by a blue star. Abbreviations: fore-aft basal length (FABL), crown height (CH), basal width (BW), anterior (ADM) and posterior denticles per millimeter (PDM). Morphometric dataset prepared as explained in methods and including measurements from DMNH 21183 reported in Supplementary Information (S2 Dataset).

Tyrannosauroidea) [78]. The Meckelian foramen (Fig 3B) is slit-like and not enlarged and rounded like in Tyrannosauridae (e.g. oral mandibular foramen in *Nanuqsaurus* [23] and foramen intramandibularis oralis in *Tyrannosaurus* [79]), a feature seen in Dromaeosauridae (e.g. *Dromaeosaurus* [80]; *Acheroraptor* [72]). The pair of anterior foramina nearby the symphysis is similar to the condition found in velociraptorines like *Acheroraptor* [72]. The shallow paradental space, a Meckelian groove that is set in the lower half of the medial side of the dentary but not directly on the ventral margin, and the hooked distal denticles of the teeth are coelurosaurian features [81, 82].

The lenticular shape of the alveoli from dorsal view (Fig 3C; S1 Fig), rather than box shaped (or squared off), is more similar to the condition in derived coelurosaurians (i.e. Maniraptora) rather than in Tyrannosauroidea [83]. The presence of distinct alveoli rather than a connected dorsal furrow excludes derived troodontids [83]. The triangular ridge on the lateral margin of the dentary (Figs 3A and 4D) is shared with *Buitreraptor* [84], *Velociraptor* (AMNH 6515) and *Tsaagan* [85], although in all these cases the presence of this structure may be an artifact of the damaged lateral rim of the dentary. The raised rim in front of the well-preserved alveolus 4 (Fig 3A) resembles the condition of the anterior alveoli in *Saurornitholestes* [43].

Interdental plates are notoriously rare in paravian theropods, and fusion or reduction of these structures has been largely considered synapomorphic for Dromaeosauridae or Deinonychosauria as a whole, except for some taxa like Microraptorinae [86], *Austroraptor* [87], *Acheroraptor* [72], and *Archaeopteryx* [88, 89]. The larger of the preserved interdental plates of DMNH 21183 is triangular (like *Acheroraptor*), with a somewhat arched apex and a broader base. The absence of medial crenulation or dorsoventral furrows in the interdental plate excludes it from assignment to Tyrannosauridae. The presence of interdental plates fused to the margin of the dentary is shared with *Atrociraptor* [90] and *Saurornitholestes* [43].

The lenticular (more labiolingually compressed than circular) cross section of the teeth in DMNH 21183 differs from the mesial and lateral dentitions of many troodontids, which have a subcircular cross-sectional outline at the crown base [83]. The teeth of DMNH 21183 lack concave surfaces adjacent to both carinae (Fig 4), as has been observed in some mesial teeth of the troodontids *Troodon* [91, 92], *Urbacodon* [93], an indeterminate troodontid taxon from Uzbekistan [93], and the troodontid *Xixiasaurus* [83, 94] as reported by [83]. As in most paravians, the teeth preserved in DMNH 21183 exhibit short interdenticular sulci (opposite of the well-developed and deep sulci present in Abelisauridae, Tyrannosauridae, and Allosauroidea [9, 83]), whereas short interdenticular sulci have been observed by Hendrickx et al. [83] in the microraptorine specimen IVPP V13476 [95], the eudromaeosaurians *Deinonychus* (YPM 5232), *Saurornitholestes* [43], and *Dromaeosaurus* (AMNH 5356; [92, 96]), as well as in some troodontids [97, 98] such as *Troodon* (NHMUK PV R.12568).

The distal denticles have a relative higher density in contrast to the much larger denticles in Troodontidae, and are more similar to the condition in Dromaeosauridae [83]. The presence of less distal denticles than mesial denticles is a more common feature found in saurornitholestine dromaeosaurids rather than Dromaeosaurinae [40]. An affinity to the sympatric cf. *Troodon* [26] can be excluded on the basis of serrated mesial teeth, which are not serrated in the troodontid [99]. An affinity with basal dromaeosaurid taxa like the halszkaraptorines and unenlagiines (e.g. *Halszkaraptor* [100], *Mahakala* [101], *Buitreraptor* [84] and *Austroraptor* [102]) can be excluded, as the dentition of these taxa are devoid of denticles [83]. Other deinonychosaurian taxa like the troodontids *Mei* [103], *Byronosaurus* [104], *Gobivenator* [105], *Urbacodon* [93], *Xixiasaurus* [94], IVPP V20378 and *Jinfengopteryx* [106], *Almas* [107, 108] and MPC-D 100–1128, the anchiornithid *Anchiornis* [109], *Eosinopteryx* [110], *Aurornis* [111], and the basal avialan *Archaeopteryx* (e.g., [112–114]) also have non-denticulate tooth crowns all along their jaws [83], excluding a potential affinity with DMNH 21183. It has to be cautioned though that independent reacquisition of denticulated teeth has been shown in some deinonychosaurian taxa like the anchiornithids *Caihong* and *Liaoningvenator* [83].

The small size of mesial denticles in DMNH 21183 is more similar to the condition in Saurornitholestinae rather than Dromaeosaurinae [43, 80, 115–118]. The shape of the denticles is slightly pointed toward the apex of the crown as in Dromaeosauridae and Troodontidae and not C- or U-shaped like in Tyrannosauridae [83]. The reduction or lack of mesial denticulation is shared with some Asian Velociraptorinae like *Tsaagan* [85]. A similar condition to DMNH 21183 where mesial teeth bear unserrated mesial carinae and denticulated distal carinae is present in many other theropod clades. This feature is also seen in the dromaeosaurid *Tsaagan* [85], the troodontids *Linhevenator* [83, 119] and possibly *Saurornithoides* (AMNH 6516 [120]).

Although hooked denticles are primarily present in some derived troodontid taxa [83, 103], there are also dromaeosaurids that have apically hooked denticles (e.g., [80, 92, 96, 121]). These include the eudromaeosaurians *Atrociraptor* and *Saurornitholestes* [83, 92, 122]. An immediately noticeable difference between denticles in Dromaeosauridae and Troodontidae is that the latter tend to bear particularly large, bulbous, and widely separated denticles, while

Dromaeosauridae have more numerous, smaller and asymmetrically convex or parallelogram-shaped denticles (Fig 4A; [83]). Other deinonychosaurian taxa lack apically hooked denticles, with a morphology that is more symmetrical and apically convex, such as in Microraptorinae and in some derived eudromaeosaurs such as *Acheroraptor*, *Bambiraptor*, *Linheraptor*, *Tsaagan*, *Utahraptor* and *Velociraptor* ([83] and references therein).

The number of denticles (12 to 13) in the partially exposed crowns of DMNH 21183 indicates that the total number of denticles per carina is greater than the ~15 present in many troodontids, particularly in taxa more derived than *Sinovenator* [26]. As Hendrickx [83] pointed out, while some tooth crowns of *Saurornitholestes* appear to have less than 15 denticles on the carina [40, 92, 123] quantitative analyses by Larson and Currie [40] indicates that the large majority of *Saurornitholestes* teeth have many more than 15 denticles on the crown (for other remarks on morphological variation in the dental series of this taxon see [43]). Within Deinonychosauria, taxa with a large number of denticles (≥6 per 1 mm) include *Richardoestesia*, Saurornitholestinae (including *Saurornitholestes*), *Sinovenator*, and *Velociraptor* [40, 83, 124–127].

While the position of the mesial margin of the teeth preserved in DMNH 21183 and the degree of surface abrasion preclude detailed morphological observation, the finely serrated mesial carina, with denticles smaller than their distal serial homologues, can be seen particularly well in the 3$^{rd}$ tooth (rdt3; Fig 3D). The presence of distal denticles larger than mesial ones was long thought to characterize the dentition of Dromaeosauridae, and this criterion was used to identify velociraptorine teeth (e.g., [83, 115–118]). Teeth with fine mesial serrations are usually characterized by a denticle size index (DSDI: the ratio between number of mesial and distal denticles) higher than 1.2 while teeth that bear carinae with subequal denticle size have usually a DSDI close to 1. These arbitrary values were proposed by Rauhut and Werner [115] and corresponds in the case of DSDI≈1.2 to approximately more than six mesial denticles for five distal serrations [83]. DMNH 21183 has a quite high DSDI (~2.3; Fig 9), which is well beyond the range of many deinonychosaurian taxa, and in the range of the most finely serrated saurornitholestine teeth (between 1 and 2.5; Fig 9). A DSDI >1.2 has been reported in the majority of eudromaeosaurians ([83] and references therein). Some lateral teeth of the troodontid *Zanabazar* [128] and some isolated crowns assigned to *Troodon* have also a very high DSDI (outliers with DSDI around 2–3; Fig 9 [40, 83, 91]. This is the opposite condition than that typically found in Dromaeosaurinae (DSDI ~ 1), like in the eponymous taxon *Dromaeosaurus* [83]. The surprisingly high DSDI of DMNH 21183 may be a juvenile trait, since many juvenile theropods have been shown to exhibit particularly fine mesial serrations relative to the distal ones, and evidence from tyrannosaurids [78, 129, 130] shows progressively decreasing DSDIs through ontogeny [135].

External textural features in dinosaur surface bone have been shown to change in relation to ontogeny [32], in a process mirroring internal microscopic remodeling [131]. For example, the skulls of ceratopsian dinosaurs show surface textural changes during ontogeny, from lightly striated to deeply rugose textures [30; 132]. Cortical bone texture with fine-grained, longitudinally striated pattern [31] is considered a size-independent criterion as an indication of relative immaturity in non-avian archosaurs [30, 31, 133]. Lightly striated cortical bone grain express nascent ontogenetic characters in theropods [31], as it is particularly clear from *Tyrannosaurus* [31], *Scipionyx* [134], and *Juravenator* [135, 136]. The same striated, fibrous bone grain texture described in relatively immature individuals of these other theropod taxa is also present in DMNH 21183 (Fig 3A and 3B). This textural feature, in combination with the diminutive size of the specimen (Table 1), is evidence that DMNH 21183 is a juvenile.

After morphological, morphometric, and phylogenetic analyses, DMNH 21183 is here interpreted as specimen of Saurornitholestinae. The recurrent clustering of DMNH 21183

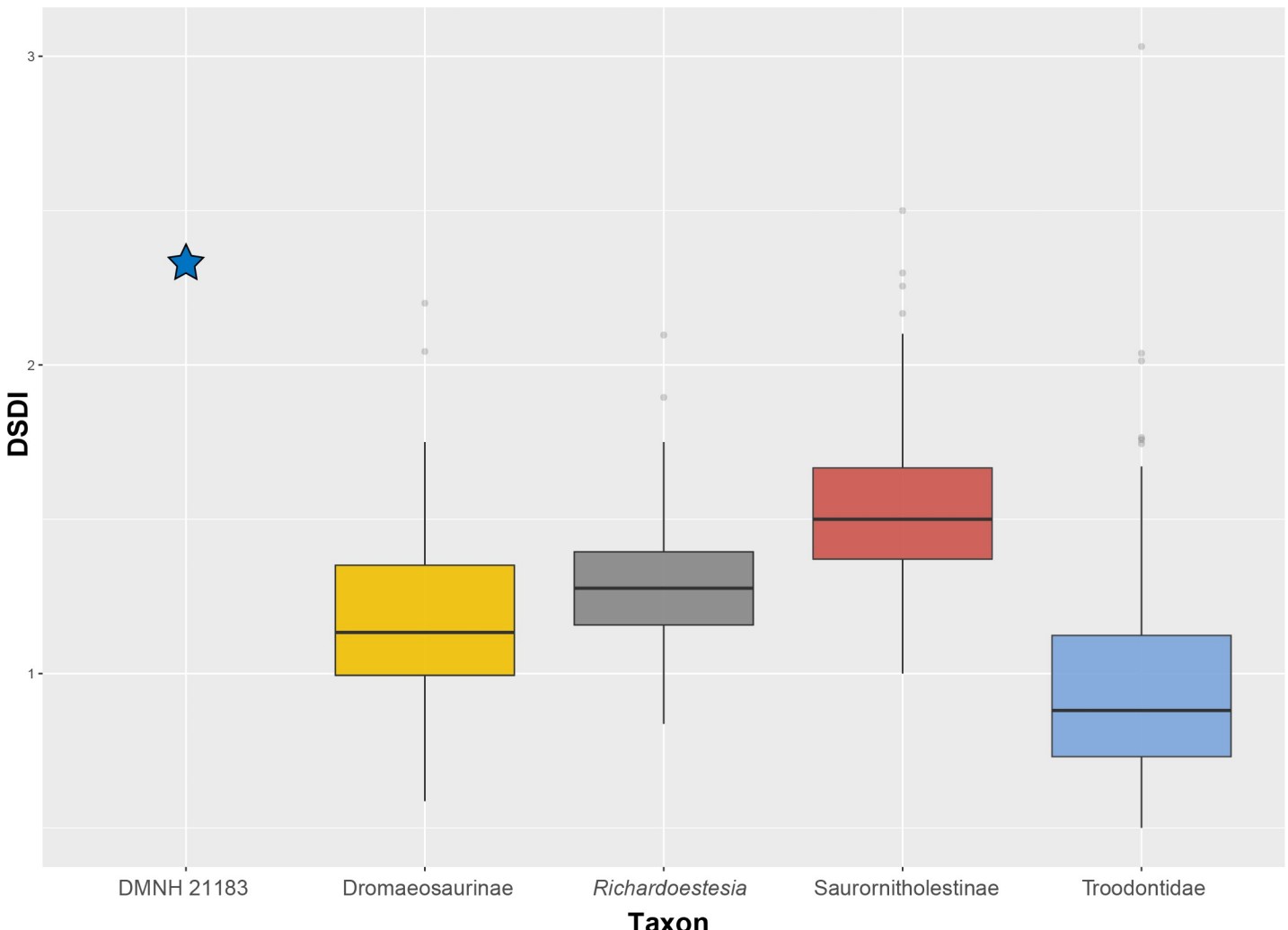

**Fig 9. Denticle size index comparison between deinonychosaurian taxa.** Box plot showing a comparison of the relative denticle size index (DSDI) between DMNH 21183 and four clades of deinonychosaurian theropods from the dataset in Larson and Currie [30]. DMNH 21183 is marked as a star. Dots represent outlier of each clade's distribution.

with Eurasian dromaeosaurids (a clade including Eudromaeosauria + Microraptorinae; Figs 5 and 6) and excluding Unenlaginae (in [33]), the sister-taxon relationship with *Saurornitholestes* (Fig 6A) in the analysis based on the dentition-only matrix from Hendrickx et al. [83], as well as in a eudromaeosaur clade more exclusive than that containing *Dromaeosaurus* in the analysis based on the tooth-crown-only matrix from Hendrickx et al. [33], all strengthen the saurornitholestine interpretation of the specimen based on morphological comparisons and teeth morphometrics.

Saurornitholestinae was first named by Longrich and Currie [14] as a subclade of Dromaeosauridae including *Saurornitholestes*, *Bambiraptor* and *Atrociraptor*. Not all phylogenetic analyses focused on deinonychosaurian interrelationships recover this clade (e.g. [101, 137]), but others do [34, 37] or partially do so [138], and we recognize this clade. Saurornitholestinae is that lineage of eudromaeosaurs closer to *Saurornitholestes* than to *Dromaeosaurus* and *Velociraptor*, and this more inclusive, stem-based concept of the name is useful when discussing isolated dental material which has often been assigned to *Saurornitholestes* or cf. *Saurornitholestes*

(e.g. [40, 96, 139]). There are several referrals to saurornitholestine dromaeosaurs in the latest Cretaceous of North America, including: the Milk River saurornitholestine [96] from the latest Santonian–earliest Campanian; the Early Campanian Menefee Formation saurornitholestine [41]; the Early Campanian Foremost Formation saurornitholestine [140]; and the Middle-Late Campanian Oldman Formation saurornitholestine [140] (S3 Dataset).

More relevant for comparative purposes to taxa in the Prince Creek Formation, *Atrociraptor marshalli* was recovered in phylogenetic analyses of paravian interrelationships as a derived member of Dromaeosauridae and close to the node comprising *Saurornitholestes* (e.g. [37, 60]). In the dental-characters-only phylogenetic analysis, DMNH 21183 clusters separately from *Atrociraptor* (Fig 6A), while the crown-only-characters and full-osteological results recover both *Atrociraptor* and DMNH 21183 in a polytomy within Eudromaeosauria (Figs 5 and 6B). There are similarities in the apically-hooked denticles on the teeth of DMNH 21183, *Saurornitholestes*, and *Atrociraptor*, but the generally larger denticles of the latter, particularly in the mesial carina [90, 141], sets it apart from *Saurornitholestes* and DMNH 21183. However, the implications of ontogenetic stage on relative denticle size should also be considered in this case. On the other hand, the interdental plates in DMNH 21183 more closely resemble those of *Atrociraptor* (TMP 95.166.1 [90]) than those of *Saurornitholestes* (TMP 1988.121.0039 [60]) in having a narrower base. *Atrociraptor* and *Saurornitholestes* are recovered in a sister-group relationship by Currie and Evans [60], and the shared similarities between DMNH 21183 and these two taxa may prove predictive should more complete dromaeosaurid material be found in the PCF.

## Paleoecological and paleobiogeographical implications of a juvenile Arctic Saurornitholestinae

The discovery of dinosaur remains at high latitudes (i.e. higher than 66˚), and in particular the abundant dinosaur bone record from the PCF, challenged traditional reptilian models for dinosaurian physiologies and inspired debate centered on the potential for long-distance migrations by dinosaurs [20, 142–144]. Given the breadth of migration patterns in extant animals (e.g. [145]), to focus our discussion it is relevant to point out that these migrations for Arctic dinosaurs were inferred to cover latitudinal distances rather than trans-Arctic migrations, which are not even observed with animals today [146]. Increased subsequent interest showed, through a variety of methods such as biomechanic, isotopic analyses, and osteohistology, that these dinosaurs likely had the necessary adaptations for overwintering in the ancient Arctic [20, 147, 148] and need not have moved to more southerly latitudes. Thus far, these discussions have focused almost exclusively on herbivorous taxa, with one exception that was based on the argument that if the dinosaurian prey did not migrate, then the predators were non-migratory as well [28].

In their review of the adaptive benefits of migration for modern mammals, Avgar and others [149] showed that in the terrestrial realm there is a decided preference for mammalian long-distance migrations to occur among large-bodied herbivores rather than carnivores. One of the suggested reasons for the rarity of migration by mammalian carnivores is that for purposes of energy consumption and mating, these animals need to establish, maintain, and defend territories. Such behavior would preclude the ability to migrate.

Dromaeosaurids were evolutionarily close to avians, and there is some discussion of the flight capabilities of some taxa within this group [150]. At best, some taxa may have been able to exhibit some rudimentary flight skills (e.g. *Microraptor* and *Zhenyuanlong* [151–153]). Further, the anatomy of most non-micraptorine dromaeosaurids, especially larger species,

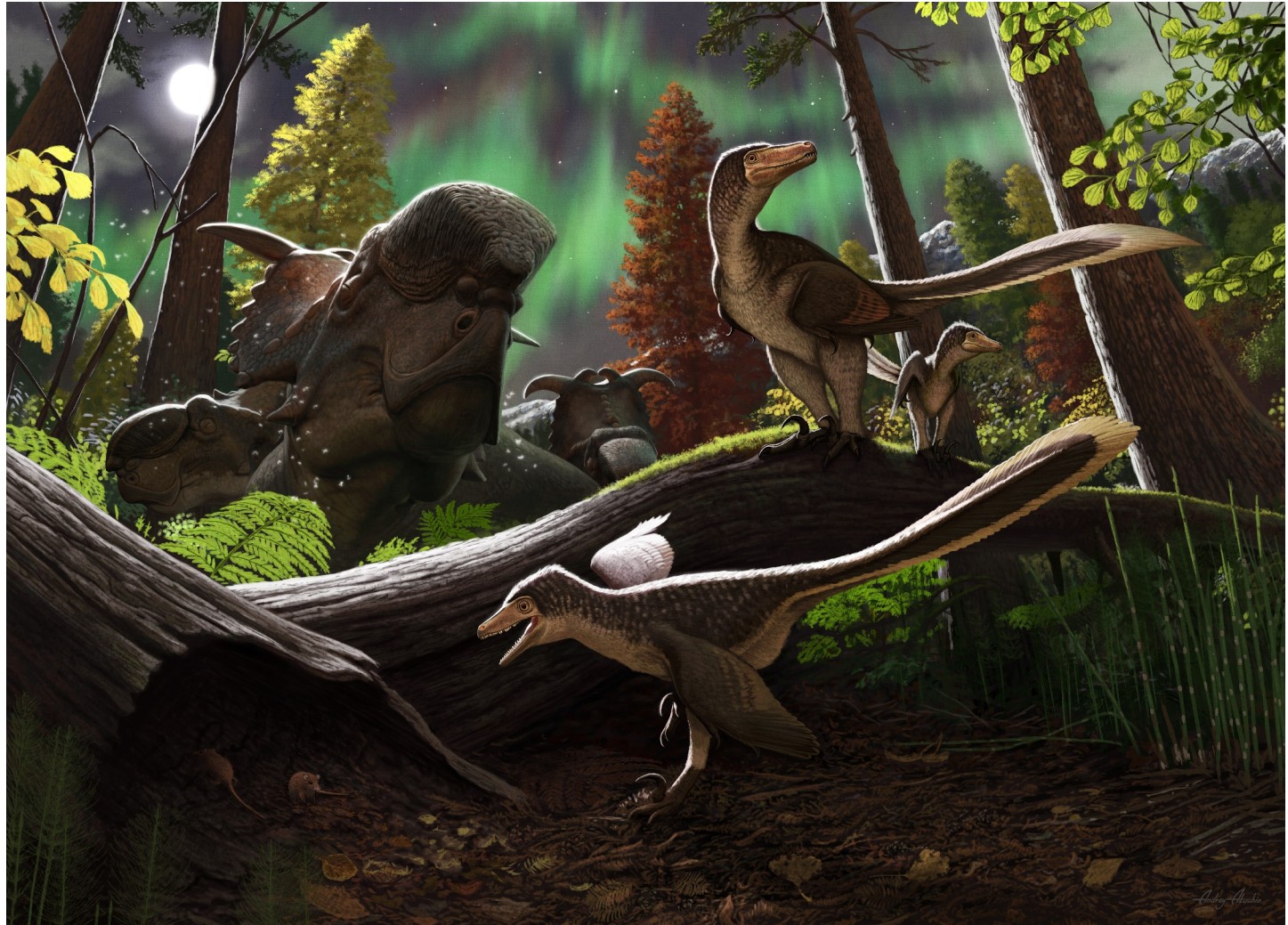

**Fig 10. Life reconstruction of the Alaskan saurornitholestine in its environment.** Artistic restoration by scientific illustrator Andrey Atuchin depicts a riparian setting in the Prince Creek Formation, matching the geological evidence described in this paper. DMNH 21183 comes from the juvenile dromaeosaurid on the branch close to the adult, while a subadult (foreground) stalks an individual of *Unnuakomys hutchisoni* [67], a methatherian known from this locality. Individuals of the sympatric ceratopsid *Pachyrhinosaurus perotorum* [22] rest in the background.

argues for a flightless lifestyle [89]. That eliminates flight as an energy efficient means for a predatory animal to cover the vast distances required to migrate across large geographic areas.

The fibrous bone texture of DMNH 21183 strongly suggests that the individual was very young at the time of its death. Given that there is also a demonstrable positive correlation between tooth size and body size, at least in some theropods (such as dromaeosaurids, but does not apply to basal ornithomimosaurs and therizinosaurs with extremely tiny teeth compared to body size [Hendrickx pers. comm. 2020]) and other diapsids [154, 155], it is reasonable to conclude that the DMNH 21183 belonged to a very small individual. Further, similar (if not stronger) biomechanical constraints for long-range migrations would apply for these small theropods, as has been advocated for ornithischian dinosaurs [156, 157]. Therefore, this specimen of a young, small individual suggests that dromaeosaurs likely nested in the ancient Arctic or in the close proximity, behavior different than long-distance migratory animals.

The identification of the first osteological remains attributed to Saurornitholestinae from the North Slope supports previous observations on the paleoecology and ecosystem structure of Late Cretaceous communities in Arctic Alaska. Previous workers suggested that gregarious herbivorous dinosaur groups inhabiting the Cretaceous Arctic were non-migratory [157]. The specimen described here (DMNH 21183) is particularly important because it represents a very small and young individual with little or limited ability to engage in long-distance travel or migrations. In modern migratory birds, it is often the case that young birds lack experience and they must rely on learning migratory habits from the adults [158]. Also, as another analog, there is a decided preference for modern mammalian long-distance migrations to occur among large-bodied herbivores rather than carnivores. Taken together, we infer that DMNH 21183 implies a perennial residency of this dromaeosaur clade (Saurornitholestinae) in the Arctic [20, 25, 26, 28]. This Alaskan Saurornitholestinae would have lived in a biotope featuring a coniferous open woodland (dominated by taxodiaceous conifers) with an angiosperm-fern understory [61, 159, 160]. Herbaceous vegetation included ferns, angiosperms, abundant horsetails and other sphenophytes [61, 159, 160]. This ancient Arctic ecosystem would have included animals such as basal ornithopods [21], the hadrosaurid *Edmontosaurus* [161], the centrosaurine *Pachyrhinosaurus* [22], the diminutive tyrannosaurid *Nanuqsaurus* [23], a large troodontid [26] and at least another dromaeosaurid taxon closer to *Dromaeosaurus* [28] than to *Saurornitholestes*. Small body-sized animals representing potential prey for the Arctic saurornitholestine (Fig 10) might have been mammals such as the methatherian *Unnuakomys* [67], a Gypsonictopidae and the multituberculate *Cimolodon* [25, 162].

DMNH 21183 adds further weight to the paleobiogeographical connection between closely related Asian and North American eudromaeosaur taxa (with sister clades present in both Asia and the Western Interior Basin of North America). DMNH 21183 is too fragmentary to provide more specific taxonomic distinction within the current record of known dromaeosaurids, but is most similar to saurornitholestines. Given the geological age of *Saurornitholestes langstoni* and *Atrociraptor marshalli*, and the wide-ranging tooth-form taxon *Richardoestesia* (Late Campanian-Late Maastrichtian), we predict that additional specimens and data may eventually provide evidence supporting the establishment of a new dromaeosaurid taxon in the Early to early Late Maastrichtian of Arctic Alaska.

## Supporting information

**S1 Fig. Magnified close-up views of the 3<sup>rd</sup> tooth in DMNH 21183.** Close-up of the mesial carina in anterior view (A, C), highlighting the denticle-bearing anterior carina (ac). Magnified lingual view of the tooth (B) highlighting the anterior (ac) and posterior (pc) carinae (D). Dotted line (C) highlights the interdenticular sulci. Scale bar: 100 μm.
(TIF)

**S2 Fig. Discriminant Functional Analysis of DMNH 21183 in Gerke and Wings [39].** Discriminant Functional Analysis of DMNH 21183 in the theropod teeth morphospace generated with the morphometric dataset provided in Gerke and Wings [39]. Abbreviations: LD, linear dimension. DMNH 21183 indicated by a green star.
(TIF)

**S3 Fig. Discriminant Functional Analysis of DMNH 21183 in Larson and Currie [40].** Discriminant Functional Analysis of DMNH 21183 in the deinonychosaurian teeth morphospace generated with the morphometric dataset provided in Larson and Currie [40]. Abbreviations: LD, linear dimension. DMNH 21183 indicated by a pink star.
(TIF)

**S4 Fig. Phylogenetic position of DMNH 21183 in Lee et al. [33].** Strict consensus topology of the shortest trees recovered by the parsimony analyses showing the position of DMNH 21183 in the matrix from Lee et al. [33] (384 MPTs, 6043 steps, CI = 0.244, RI = 0.587). Numbers adjacent to nodes are the bootstrap values. Red box highlights the node containing DMNH 21183 (red arrow).
(TIF)

**S5 Fig. Phylogenetic position of DMNH 21183 in Hendrickx et al. [35].** Strict consensus topology of the shortest trees recovered by the parsimony analyses showing the position of DMNH 21183 in the dentition-only character matrix from Hendrickx et al. [35] (2 MPTs, 1314 steps, CI = 0.194, RI = 0.418). The overall topology was constrained with DMNH 21183 allowed to float. Numbers adjacent to nodes are the bootstrap values. Red box highlights the node containing DMNH 21183 (red arrow) in Eudromaeosauria.
(TIF)

**S6 Fig. Phylogenetic position of DMNH 21183 in Hendrickx et al. [35].** Strict consensus topology of the shortest trees recovered by the parsimony analyses showing the position of DMNH 21183 in the tooth-crown-only character matrix from Hendrickx et al. [35] (5 MPTs, 867 steps, CI = 0.183, RI = 0.439). The overall topology was constrained with DMNH 21183 allowed to float. Numbers adjacent to nodes are the bootstrap values. Red box highlights the more inclusive node containing DMNH 21183 (red arrow) in Eudromaeosauria.
(TIF)

**S1 Table. Systematic definitions used in this study.** Systematic names and relative phylogenetic definition used in this study.
(XLSX)

**S1 Dataset. Phylogenetic character scoring of DMNH 21183.** Character scoring in the phylogenetic matrices from Lee et al. [33] and Hendrickx et al. [35].
(RTF)

**S2 Dataset. Morphometric data.** Morphometric scoring for DMNH 21183 and modified datasets for multivariate analyses (PCA and DFA) in Gerke and Wings [39] and Larson and Currie [40]. Systematic entries follow methodology as described in the Material and Methods section. Original datasets with specimen-level denominations can be found in Gerke and Wings (https://doi.org/10.1371/journal.pone.0158334.s001) and Larson and Currie (https://doi.org/10.1371/journal.pone.0054329.s001).
(XLSX)

**S3 Dataset. Latest Cretaceous dromaeosaurids in North America.** Faunal list compilation of all dromaeosaurid taxa in the latest Cretaceous (84.5–66.043 million years ago) with geographic, stratigraphic, chronological, and literature information attached.
(XLSX)

## Acknowledgments

Andrey Atuchin is acknowledged for the commissioned reconstruction in Fig 10. Thomas Carr (Carthage College, USA) and Christophe Hendrickx (Unidad Ejecutora Lillo, CONICET-Fundación Miguel Lillo, Argentina) are also thanked for their thorough reviews which greatly improved the quality of this manuscript. We acknowledge Renata Tully (Nikon, Inc.) and Ryan Clubb (Keyence Corp. of America) for microscope and software support. The phylogenetic software TNT is available through the supporting and sponsorship of the Willi Hennig

Society. Lastly, the Arctic Management Unit of the Bureau of Land Management provided administrative support. The specimen discussed here was collected under BLM permit number AA-86864.

## Author Contributions

**Conceptualization:** Alfio Alessandro Chiarenza, Anthony R. Fiorillo.

**Data curation:** Alfio Alessandro Chiarenza, Anthony R. Fiorillo, Ronald S. Tykoski, Paul J. McCarthy, Peter P. Flaig.

**Formal analysis:** Alfio Alessandro Chiarenza, Paul J. McCarthy, Peter P. Flaig.

**Funding acquisition:** Anthony R. Fiorillo.

**Investigation:** Alfio Alessandro Chiarenza, Ronald S. Tykoski, Paul J. McCarthy, Peter P. Flaig.

**Methodology:** Alfio Alessandro Chiarenza, Dori L. Contreras.

**Supervision:** Alfio Alessandro Chiarenza, Anthony R. Fiorillo.

**Validation:** Alfio Alessandro Chiarenza, Ronald S. Tykoski.

**Visualization:** Alfio Alessandro Chiarenza, Ronald S. Tykoski.

**Writing – original draft:** Alfio Alessandro Chiarenza.

**Writing – review & editing:** Alfio Alessandro Chiarenza, Anthony R. Fiorillo, Ronald S. Tykoski, Paul J. McCarthy, Peter P. Flaig, Dori L. Contreras.

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
