## [Decision Letter · Decision Letter 0]

8 Apr 2020

PONE-D-20-05495

The first juvenile dromaeosaurid (Dinosauria: Theropoda) from Arctic Alaska

PLOS ONE

Dear Dr. Chiarenza,

Thank you for submitting your manuscript to PLOS ONE. After careful consideration, we feel that it has merit but does not fully meet PLOS ONE’s publication criteria as it currently stands. Therefore, we invite you to submit a revised version of the manuscript that addresses the points raised during the review process.

Both reviewers suggested moderate revisions to the manuscript. Please read and carefully consider the reviewers' comments below, and address their concerns and suggestions in your revised submission.

We would appreciate receiving your revised manuscript by May 16 2020 11:59PM. To enhance the reproducibility of your results, we recommend that if applicable you deposit your laboratory protocols in protocols.io, where a protocol can be assigned its own identifier (DOI) such that it can be cited independently in the future. For instructions see: http://journals.plos.org/plosone/s/submission-guidelines#loc-laboratory-protocols

We look forward to receiving your revised manuscript.

Kind regards,

Laura Beatriz Porro, Ph.D.

Academic Editor

PLOS ONE

Journal Requirements:

2. We note that Figure 1 in your submission contain [map/satellite] images which may be copyrighted. All PLOS content is published under the Creative Commons Attribution License (CC BY 4.0), which means that the manuscript, images, and Supporting Information files will be freely available online, and any third party is permitted to access, download, copy, distribute, and use these materials in any way, even commercially, with proper attribution. For these reasons, we cannot publish previously copyrighted maps or satellite images created using proprietary data, such as Google software (Google Maps, Street View, and Earth). For more information, see our copyright guidelines: http://journals.plos.org/plosone/s/licenses-and-copyright.

You may seek permission from the original copyright holder of Figure(s) [#] to publish the content specifically under the CC BY 4.0 license. 

If you are unable to obtain permission from the original copyright holder to publish these figures under the CC BY 4.0 license or if the copyright holder’s requirements are incompatible with the CC BY 4.0 license, please either i) remove the figure or ii) supply a replacement figure that complies with the CC BY 4.0 license. Please check copyright information on all replacement figures and update the figure caption with source information. If applicable, please specify in the figure caption text when a figure is similar but not identical to the original image and is therefore for illustrative purposes only.

Reviewers' comments:

Reviewer's Responses to Questions

**Comments to the Author**

1. Is the manuscript technically sound, and do the data support the conclusions?

Reviewer #1: Yes

Reviewer #2: Partly

2. Has the statistical analysis been performed appropriately and rigorously? 

Reviewer #1: Yes

Reviewer #2: I Don't Know

3. Have the authors made all data underlying the findings in their manuscript fully available?

Reviewer #1: Yes

Reviewer #2: Yes

4. Is the manuscript presented in an intelligible fashion and written in standard English?

Reviewer #1: Yes

Reviewer #2: No

5. Review Comments to the Author

Reviewer #1: In their manuscript entitled “The first juvenile dromaeosaurid (Dinosauria: Theropoda) from Arctic Alaska”, Chiarenza and colleagues report, describe and identify the anterior portion of a dentary (DMNH 21183) from the Early Maastrichtian Prince Creek Formation of northern Alaska. The specimen is referred to a juvenile saurornitholestine dromaeosaurid based on comparative anatomy and a morphometric analysis using the best preserved tooth (d3), an identification I agree with. This paper is well-written, well-organized and most of the figures are very nice. The descriptive part is thorough and the introduction, geological settings and discussion section are sound and detailed. The methodology used to identify the material could certainly be improved though, and I have some problems with the way the text discussing the phylogenetic distribution of the material based on dental features is written. If I believe that this contribution only provides a limited amount of information for our knowledge on theropods/paravian paleogeography and palaeoecology, I would nonetheless suggest to accept Chiarenza et al.’s work with moderate revisions, urging the authors to take into consideration my main recommendations:

- My main problem with this paper lies in the author’s method to identify the material. First, I wonder why they did not perform a cladistic analysis using the most recent datamatrix on coelurosaurs/paravian relationships. I understand that their material is very limited but wonder if it does not display enough apomorphic features to help resolve its phylogenetic affinity. At least, the authors should specify why they do not want to conduct a cladistic analysis to explore the phylogenetic distribution of DMNH 21183. Second, they decide to perform a PCA based on crown-measurements of d3 to help in the identification of the material. Yet, I believe there are more appropriate methods to identify theropod teeth such as discriminant and cluster analyses using a dataset of crown-based measurements, or (even better according to me) a cladistic analysis using a dentition-based datamatrix and a fully constrained topological tree (see my paper on the Aerosteon tooth and the dentition of Abelisauridae recently published in Cretaceous Research on how to perform those analyses). That being said, the major problem I see is that the best preserved tooth of this small portion of mandible, d3, is a partially erupted tooth and I wonder how the authors managed to accurately take measurements of FABL, CH and BW on this crown. In addition, it does not make much sense to me to use a dataset restricted to paravians and to omit specimens from which the mesial carina does not include denticles. I would therefore recommend the authors to perform a first analysis with all taxa and measurements from Larson & Currie’s dataset, then use a more restrictive one based on relevant arguments on why tyrannosauroids, for instance, should be excluded. I would also strongly advise them to perform discriminant and cluster analyses on the dataset of crown-based measurements and, ideally, a cladistic analysis using my most recent dentition-based datamatrix (from the Aerosteon paper), coding their dental material as coming from the dentary and the mesial dentition. I am quite confident that the result of the cladistic analysis would further support a saurornitholestine affinity.

- In the methodology section, the authors should specify the result(s) of the cladistic analysis of what publications they follow when commenting on paravian phylogeny and the phylogenetic position of the different dromaeosaurids. Likewise, they should add the fact that it is the third dentary tooth of DMNH 21183 that is included in the PCA.

- In the chapter on morphological remarks (which I would suggest to rename “Phylogenetic position of DMNH 21183”), in the section in which the authors discuss the distribution of dental features displayed by DMNH 21183, they obviously use my recent paper on the distribution of dental features in theropods (Palaeontologica Electronica) as a main reference. I thank them for that but, if my paper is often cited, it frustrates me to see that some parts of the text directly comes from my paper without citing it. In many parts of the text, the authors just copied and pasted portions of my PE publication, using the exact same citations and specimen numbers, while I highly doubt that they have personally examined the material I have, or looked at the references the way I have. For fairness (and because sometimes it almost looks like plagiarism), I would really like the authors to rephrase the text that directly comes from my paper using the later as the main bibliographical source, and remove all specimens (those coming directly from my paper) that they did not examine themselves. In this section of the MS, the DSDI for crowns having mesial and distal roughly of the same size is also close to 1 and not lower than 0.9.

- I don’t have any comments to make regarding the Introduction, Geological setting and Depositional Environments, and Discussion sections, which really seem to be sound and well-written. I would only recommend to move the part of the text that discuss the juvenile features displayed by the specimen, present in the description section, to the Results or Discussion sections (I always prefer discussing the affinities of a specimen and its ontogenetic stage in the Discussion section, personally).

- Regarding figures, I would only suggest the authors to make the convex hull and the arrows of the variables used in the PCA better visible to the reader, in figure 6. Many taxa must also be italicized, while the capital letters of some titles of articles should be removed, in the bibliography section.

I provided additional corrections, remarks and suggestion in the pdf version of their MS, and ask the authors to take them into consideration.

Kind regards,

Christophe Hendrickx, San Miguel de Tucumán, the 14th of March 2020.

Reviewer #2: March 30, 2020

Dear Editor,

I submit to you my review of PONE-D-20-05495, “The first juvenile dromaeosaurid (Dinosauria: Theropoda) from Arctic Alaska” by Chiarenza et al. The authors report on the first skeletal material of a dromaeosaurid from the Arctic, which is an important data point; fortunately, the specimen, although fragmentary, has some important features that enables comparisons with other taxa. I think that the article is publishable with minor revisions. Most of the fixes center on writing style.

In terms of major concerns, I find the second paragraph of the Discussion to be somewhat convoluted and difficult to follow; I think it can be redrafted into a more straightforward style so that the main points aren’t lost to the reader.

The third paragraph makes the claim that the Arctic saurornitholestines were nonmigratory; in my view, the evidence is mute on that point and the authors should present a more critical assessment of such claims; for example, what evidence is required to support the claim? Is that evidence available based on the fossil at hand, really?

Finally, the fourth paragraph continues the argument, but from what strikes me as – at best - tangential lines of evidence. Again, I encourage the authors to re-visit this section with a critical bent of mind. Other comments on the discussion appear in the specific comments below.

Unless I missed it, there is one significant gap in the discussion, and that is comparison with the dromaeosaurid fauna of the Horseshoe Canyon Formation of Alberta, which is equivalent in geological age to the PCF. I suggest that the authors make an explicit comparison with the HCF dromaeosaurids to arrive at a snapshot of the distribution of the clade at that time in the northern half of Laurasia. I think that is the opportunity that will lock in future citations of the article, once it is published.

Finally, the authors identify a feature (lateral ridge of the dentary) in the new fossil that is also only seen elsewhere in the Gondwanan Buitreraptor, but later in the ms, they consider the fossil referable to Saurornitholestinae based on a morphometric analysis of teeth. I ask that the authors acknowledge these conflicting hypotheses of identity and to explicitly provide their rationale for preferring one hypothesis over the other, if that is what they decide, in the end, to do.

My statistical experience does not extend to PCA analyses, so I am unable to comment on that part of their work.

The authors may know my identity: Thomas D. Carr

Specific comments:

Line 17: delete “the”; taxonomic names are formal names and so are not preceded by “the.”

Line 19: replace “diminutive” with “small”; avoid jargon.

Line 20: delete “morphological”, replace “based on” with “with.”

Line 23: replace “status” with “growth stage”; as far as I can tell relative maturity isn’t a status. Replace “exploration” with “comparison.”

Line 62: Replace “taxonomical” with “taxonomic.”

Line 65: delete “purported”, unless you are doubtful of its uniqueness.

Line 95: replace “like” with “such as.”

Line 104: replace “examined” with “studied first hand” and modify the sentence accordingly.

Line 106: replace “performed” with “did.”

Line 107: insert a comma after “measurements”; find an accurate replacement for “distilling”.

Line 108: replace “representing” with “that represents”; delete “to be.”

Line 110: replace “estimate…of” with “compare”; replace “within” with “with.”

Line 208: is it “foramen” or “foramina”; the figure labels only one foramen. Please clarify.

Line 211: is there any evidence for the first alveolus? How do you justify the identification of the first preserved tooth as the second tooth in sequence? Clarify.

Line 215: replace “shows” with “has.”

Line 216: delete “of…tooth.”

Line 217: replace “dividing” with “separating.”

Line 222: rephrase, “slightly anteroposteriorly longer than mediolaterally wide.”

Line 224: replace “mid-way depth” with “midheight.”

Line 225: replace “a” with “an.”

Line 229: replace “, but” with “that is”; the word “but” is used to introduce an exception or contradiction, not a similarity or an elaboration upon a point.

Line 231: replace “mesodistal” with “mesiodistal.”

Line 233: replace “damaging” with “damage.”

Line 235: insert “13” ahead of “denticles.”

Line 236: delete “, with 13 denticles.”

Line 239: word choice – does “coarser” mean “larger”? If so, say so to remove the ambiguity.

Line 242: replace “observed” with “seen.”

Line 244: replace “a hypothetical” with “an”; how was this estimate arrived at? Briefly justify.

Line 246: delete “morphological.”

Line 253: what, exactly, is rounded and not pointed? I suspect this is a grammatical issue; please fix.

Line 263: replace “grains” with “grain.”

Line 266. Replace ”in” with “along with.”

Line 267: replace “points…21183” with “is evidence that DMNH 21183 is a juvenile.” Avoid passive voice – in science, it is ok to be declarative! In contrast, passive voice gives the impression that you are backing away from your evidence, which defeats the purpose of your hard work.

Line 274: delete “traced and.”

Line 294: delete “combination.”

Lines 294,295: “oval” and “lenticular” mean the same thing; pick one to avoid redundancy.

Lines 300,301: replace “relatively better” with “well.”

Line 310: replace “between” with “among.”

Lines 315,316: how is it that you can comment on the form of the first dentary tooth after you’ve made it clear that it is completely missing? Clarify.

Line 324: replace “observed” with “reported.”

Line 328: replace “with” with “have”; replace “compared” with “in contrast.” Also, does “coarser” just mean “larger?” Clarify.

Line 333: replace “this” with “the”; delete “taxon.”

Line 335: what about the ridge?!

Line 339: make sure that “anchiornithines” is the correct moniker to use. I.e., is there an “Anchiornithinae”?

Line 343: replace “It….that” with “However,”; replace “been shown” with “occurred.”

Line 345: pluralize “anchiornithid.”

Lines 245 to 346: delete “, in….Eudromaeosauria.”

Line 349: replace “reduction in” with “small” since there is no evolutionary or developmental process described here – you really don’t know if the size is plesiomorphic or the juvenile condition; avoid process-based terms when describing static, context-free morphology.

Line 353: replace “recalls a similar” with “is similar to the.”

Line 354: pluralize “Velociraptorinae.”

Line 357: “in-serrated/absent” are equivalent terms – pick one to reduce the annoyance of your reader. Concision is always appreciated in heavy osteodental descriptions!

Line 358: replace “observed” with “also seen in”; again, avoid passive voice.

Line 365: replace “exhibit” with “have.”

Line 387: replace “are” with “include.’

Line 392: delete “of the dentary.” Avoid redundancy.

Line 394: replace “therefore” with “this criterion was”; delete “as…feature.”

Line 395: delete “versus Dromaeosaurinae.”

Line 397: briefly define DSDI/

Line 402: is there a citation for the saurornitholestine teeth? What is their value?

Lines 410-411: move these up to line 402 so that your reader can see how the saurornitholestine values compare right away. By the way, is there a difference between Saurornitholestinae and Velociraptorinae? Is see that Saurornitholestes is a velociraptorine, so Velociraptor is not a saurornitholestine?

Line 414: what is the DSDI? Give the value!

Lines 419 to 422: methods are best written in past tense. Please fix.

Line 424: replace “provided” with “given.”

Lines 434 to 437: fix the commas.

Line 436: replace “provided” with” represented.”

Line 438: replace “presented” with “given.”

Lines 444 to 446: didn’t you earlier state that the lateral ridge allies the specimen with Buitreraptor? How do you weight the dental data against the osteological evidence? Justify, especially since you are dealing with a juvenile specimen.

Line 449: please stop using “status” for “growth stage.”

Lines 480 to 482: Given that dromaeosaurids are part of a clade of flying theropods, why couldn’t have they just flown along their migratory routes? Explain.

Line 488: replace “southerner” with “southern.”

Lines 485 to 488: why is large size evidence for “success”? Justify. What is meant by “success”? Clarify.

Line 493: change “stronger constrained” to “greater constraints.” What sort of constraints? Clarify.

Lines 494 to 495: convoluted; fix.

Line 498: “eggs” should be singular.

Lines 494 to 505: perhaps it is because it is late in the day, but I find this section to be very difficult to follow. Please redraft with a clearer argument and straightforward sentence structure.

Line 506: avoid passive voice; fix.

Lines 510 to 512: this can be shortened significantly; fix.

Lines 513 to 515: the mere presence of taxon does not imply migration any more than it doesn’t imply migration. Fix.

Line 523: should there be an “r” in “cfr”? Isn’t it just “cf.”? Where’s the period? Fix.

Line 525: replace “appear” with “are.”

Line 526: replace “but” with “and” since you aren’t marking a difference or exception.

Line 544: a period should follow “al”; e.g., Osborn et al. (1905); replace “for” with “that.”

Line 545: insert “is” after the taxon name.

Lines 513 to 555: I find this section to be, uncomfortably, speculative, but I leave it up to the judgment of the author and co-authors whether or not it needs to be reined in. Perhaps more citations in this section would put any concerns to rest. It just strikes me as a bridge too far when all of it is merely based on the relative abundance of teeth and – as far as I can tell - inconclusive functional inferences of tooth morphology. Please reconsider this section in a more critical light.

Lines 560 to 562: You don’t really know that it wasn’t migratory; it could be that all small dromaeosaurids were fully capable of flight early in growth. Please reconsider this point more critically – does the inference really have support?

Lines 564 to 565: again, you have no rationale to think this – if dromaeosaurids were volant, there’s literally nothing to stop them from migrating. It’s possible flight brought deinonychosaurians their global distribution. Regardless, I don’t think you have the evidence to make the claim, unless there’s something obvious that I missed earlier.

Figure 3: “a3” is not labeled; label both interdental plates; label the circular fossa.

6. PLOS authors have the option to publish the peer review history of their article (what does this mean?). If published, this will include your full peer review and any attached files.

Reviewer #1: Yes: Christophe Hendrickx

Reviewer #2: Yes: Thomas D. Carr

---

## [Author Response · Author response to Decision Letter 0]

11 May 2020

Dear Editor, 

Thanks for editing our manuscript entitled “The first juvenile dromaeosaurid (Dinosauria: Theropoda) from Arctic Alaska”. Attached to this letter we provide a revised version of the Manuscript with updated figures and Supplementary material. Given the comments from the reviewers, we accommodated their requests by: 1) adding a phylogenetic systematics section; 2) expanding the morphometric dataset with a more inclusive group of theropod taxa for additional analysis, adding also discriminant function analyses to the pool of analytical tools employed in the study; 3) streamling and partially reworking the discussion, broadening the section with more paleoecological and biogeographic insights but focusing the section. We added an additional author, Dr. Dori Contreras to the team behind the study: other than offering additional insights throughout the manuscript and assisting with the phylogenetic analysis, she provided important paleobotanical information for the new Figure 10. We strongly appreciated the input from the reviewers, which greatly improved the quality of this manuscript. All figures in this paper have been firsthand produced by our authorial team, including the map and panoramic view in Figure 1 which has not been produced with the aid of any external software (e.g. Google Maps) but has been photographed and assembled by one of our coauthors (Paul J. McCarthy).The artistic reconstruction in Figure 10 was commissioned by our team to scientific illustrator Andrey Atuchin, hence we own the rights to the illustration.

Below is outlined a point by point response to both Reviewers. Whereas their text has been highlighted in italics, our response has been written in bold (see attached word document for this formating).

Best Regards,

Alfio Alessandro Chiarenza

On behalf of all coauthors.

 

5. Review Comments to the Author

Reviewer #1: In their manuscript entitled “The first juvenile dromaeosaurid (Dinosauria: Theropoda) from Arctic Alaska”, Chiarenza and colleagues report, describe and identify the anterior portion of a dentary (DMNH 21183) from the Early Maastrichtian Prince Creek Formation of northern Alaska. The specimen is referred to a juvenile saurornitholestine dromaeosaurid based on comparative anatomy and a morphometric analysis using the best preserved tooth (d3), an identification I agree with. This paper is well-written, well-organized and most of the figures are very nice. The descriptive part is thorough and the introduction, geological settings and discussion section are sound and detailed. The methodology used to identify the material could certainly be improved though, and I have some problems with the way the text discussing the phylogenetic distribution of the material based on dental features is written. If I believe that this contribution only provides a limited amount of information for our knowledge on theropods/paravian paleogeography and palaeoecology, I would nonetheless suggest to accept Chiarenza et al.’s work with moderate revisions, urging the authors to take into consideration my main recommendations:

- My main problem with this paper lies in the author’s method to identify the material. First, I wonder why they did not perform a cladistic analysis using the most recent datamatrix on coelurosaurs/paravian relationships. I understand that their material is very limited but wonder if it does not display enough apomorphic features to help resolve its phylogenetic affinity. At least, the authors should specify why they do not want to conduct a cladistic analysis to explore the phylogenetic distribution of DMNH 21183. Second, they decide to perform a PCA based on crown-measurements of d3 to help in the identification of the material. Yet, I believe there are more appropriate methods to identify theropod teeth such as discriminant and cluster analyses using a dataset of crown-based measurements, or (even better according to me) a cladistic analysis using a dentition-based datamatrix and a fully constrained topological tree (see my paper on the Aerosteon tooth and the dentition of Abelisauridae recently published in Cretaceous Research on how to perform those analyses). That being said, the major problem I see is that the best preserved tooth of this small portion of mandible, d3, is a partially erupted tooth and I wonder how the authors managed to accurately take measurements of FABL, CH and BW on this crown. In addition, it does not make much sense to me to use a dataset restricted to paravians and to omit specimens from which the mesial carina does not include denticles. I would therefore recommend the authors to perform a first analysis with all taxa and measurements from Larson & Currie’s dataset, then use a more restrictive one based on relevant arguments on why tyrannosauroids, for instance, should be excluded. I would also strongly advise them to perform discriminant and cluster analyses on the dataset of crown-based measurements and, ideally, a cladistic analysis using my most recent dentition-based datamatrix (from the Aerosteon paper), coding their dental material as coming from the dentary and the mesial dentition. I am quite confident that the result of the cladistic analysis would further support a saurornitholestine affinity.

We appreciate the positive comments from the Reviewer. We decided to largely follow his recommendation in proceeding with a set of phylogenetic tests before a quantitative-morphometric analysis of the material. In this resubmission we first included DMNH 21183 in the phylogenetic dataset from Lee et al. (2014), a dataset with a wide taxonomic sampling but also with a large pool of dentary characters to maximize the scorings for our fragmentary specimen. We then included the dental coding from the specimen in the dentition-only and crown-tooth-only data matrices from Hendrickx et al. (2020), an additional set of analyses which strengthen our previous interpretation based on comparative anatomy and morphometrics. We also embraced the recommendations related to morphometric by first performing multivariate analyses on the teeth morphometric dataset by Gerke and Wings (2016) and then moving on the less inclusive Larson and Currie (2016). We want to point out that the choice of Gerke and Wings (2016) was based on the need for a purely morphometric (that is quantitative) dataset to compare our specimen, as we wanted to keep consistency with the measurements from Larson and Currie (2013) but also wanted to have an independent test which could exclude any qualitative observations (making it fully independent from the comparative and phylogenetic results). We strongly acknowledge the reviewer in his suggestions, as these additional analyses not only strengthened our interpretation, but also provided scope for expanded discussion on this specimen.

- In the methodology section, the authors should specify the result(s) of the cladistic analysis of what publications they follow when commenting on paravian phylogeny and the phylogenetic position of the different dromaeosaurids. Likewise, they should add the fact that it is the third dentary tooth of DMNH 21183 that is included in the PCA.

We thank the Reviewer for this suggestion, this is now been included in the Material and Methods section.

- In the chapter on morphological remarks (which I would suggest to rename “Phylogenetic position of DMNH 21183”), in the section in which the authors discuss the distribution of dental features displayed by DMNH 21183, they obviously use my recent paper on the distribution of dental features in theropods (Palaeontologica Electronica) as a main reference. I thank them for that but, if my paper is often cited, it frustrates me to see that some parts of the text directly comes from my paper without citing it. In many parts of the text, the authors just copied and pasted portions of my PE publication, using the exact same citations and specimen numbers, while I highly doubt that they have personally examined the material I have, or looked at the references the way I have. For fairness (and because sometimes it almost looks like plagiarism), I would really like the authors to rephrase the text that directly comes from my paper using the later as the main bibliographical source, and remove all specimens (those coming directly from my paper) that they did not examine themselves. In this section of the MS, the DSDI for crowns having mesial and distal roughly of the same size is also close to 1 and not lower than 0.9.

We thank the reviewer for these suggestions: we proceeded in renaming the section, paraphrasing further the content referring to his work, adding additional citations to them and reporting all of his edits (e.g. related to the DSDI of some taxa). We hope that this resubmission fully reflects the credit due to his scientific production.

- I don’t have any comments to make regarding the Introduction, Geological setting and Depositional Environments, and Discussion sections, which really seem to be sound and well-written. I would only recommend to move the part of the text that discuss the juvenile features displayed by the specimen, present in the description section, to the Results or Discussion sections (I always prefer discussing the affinities of a specimen and its ontogenetic stage in the Discussion section, personally).

Thanks for the appreciative comments. We also agreed with the recommendation regarding the rearrangement of the text and proceeded doing so in this resubmission. 

- Regarding figures, I would only suggest the authors to make the convex hull and the arrows of the variables used in the PCA better visible to the reader, in figure 6. Many taxa must also be italicized, while the capital letters of some titles of articles should be removed, in the bibliography section.

Thanks for these observations, we modified these figures and followed the same for the new ones added to this resubmission.

I provided additional corrections, remarks and suggestion in the pdf version of their MS, and ask the authors to take them into consideration.

We thank the reviewer for these thorough edits. We accepted and modified all relevant portions of the main manuscript text accordingly.

Kind regards,

Christophe Hendrickx, San Miguel de Tucumán, the 14th of March 2020.

Reviewer #2: March 30, 2020

Dear Editor,

I submit to you my review of PONE-D-20-05495, “The first juvenile dromaeosaurid (Dinosauria: Theropoda) from Arctic Alaska” by Chiarenza et al. The authors report on the first skeletal material of a dromaeosaurid from the Arctic, which is an important data point; fortunately, the specimen, although fragmentary, has some important features that enables comparisons with other taxa. I think that the article is publishable with minor revisions. Most of the fixes center on writing style.

In terms of major concerns, I find the second paragraph of the Discussion to be somewhat convoluted and difficult to follow; I think it can be redrafted into a more straightforward style so that the main points aren’t lost to the reader.

The third paragraph makes the claim that the Arctic saurornitholestines were nonmigratory; in my view, the evidence is mute on that point and the authors should present a more critical assessment of such claims; for example, what evidence is required to support the claim? Is that evidence available based on the fossil at hand, really?

Finally, the fourth paragraph continues the argument, but from what strikes me as – at best - tangential lines of evidence. Again, I encourage the authors to re-visit this section with a critical bent of mind. Other comments on the discussion appear in the specific comments below.

Unless I missed it, there is one significant gap in the discussion, and that is comparison with the dromaeosaurid fauna of the Horseshoe Canyon Formation of Alberta, which is equivalent in geological age to the PCF. I suggest that the authors make an explicit comparison with the HCF dromaeosaurids to arrive at a snapshot of the distribution of the clade at that time in the northern half of Laurasia. I think that is the opportunity that will lock in future citations of the article, once it is published.

Finally, the authors identify a feature (lateral ridge of the dentary) in the new fossil that is also only seen elsewhere in the Gondwanan Buitreraptor, but later in the ms, they consider the fossil referable to Saurornitholestinae based on a morphometric analysis of teeth. I ask that the authors acknowledge these conflicting hypotheses of identity and to explicitly provide their rationale for preferring one hypothesis over the other, if that is what they decide, in the end, to do.

My statistical experience does not extend to PCA analyses, so I am unable to comment on that part of their work.

The authors may know my identity: Thomas D. Carr

We would thank the Reviewer for his suggestions. We proceeded in this resubmission with a double check in writing style and grammar. Given also the recommendations from Reviewer 1, we decided to expand the study with additional phylogenetic and morphometric analyses. As a consequence, the Discussion section has been now partially reworded and reorganized, hoping to make its style more readable by the audience. We appreciate the suggestion to expand our discussion on paleoecology and biogeography of other contemporary dromaeosaurids. We did so not only by expanding the relevant discussion paragraphs, but also reporting some additional data (Dataset S3) to compare more easily the temporal and geographic distribution of North American dromaeosaurids during the latest Cretaceous. Regarding the anatomical observations, after we expanded our search because of the suggestions from Reviewer 1, we found that the structure we previously considered a potential synapomorphy between Buitreraptor and DMNH 21183 as to be more widespread in Dromaeosauridae. In particular we noticed the presence of similar lateral ridges in many other dentary rims of dromaeosaurids (e.g. Velociraptor AMNH 6515 right side, Tsaagan IGM 100/1015 right side) which are partially emphasized by damaging of the bone surface. We also want to remark that the absence of denticulation in Unenlaginae (as reported in the former draft of the paper) already excludes the identification of DMNH 21183 as a closer taxon to, for example Buitreraptor, than to more derived dromaeosaurids (e.g. Eudromaeosauria). We then modified the text accordingly to reflect these observations. We also added more osteological information that popped up while scoring the specimen for phylogenetic analyses, strengthening our previous interpretation based on comparative anatomy. We are also sure that the reviewer will appreciate the efforts in providing a suite of phylogenetic tests to our hypotheses, supporting our previous interpretation related to the systematic position of this Alaskan theropod.

Lastly we expanded the section related to the paleoecology of Arctic dromaeosaurs which the reviewer previously considered speculative. We provided more context based particularly on comparison with modern theropods’ migratory habits in the Arctic region. We also added information from the literature on growth rates and other indicative physiological observations that would back up our interpretation that these Arctic theropods indeed would spend their full solar year in the Arctic without migrating to southern latitudes. We removed, as the Reviewer recommended, some more speculative ecomorphological content, in order to simplify and focus further our Discussion section. We furthermore appreciated the effort in proof-reading our manuscript, and provided below a point by point check of the edits added to this resubmission.

Specific comments:

Line 17: delete “the”; taxonomic names are formal names and so are not preceded by “the.”

Done

Line 19: replace “diminutive” with “small”; avoid jargon.

Done.

Line 20: delete “morphological”, replace “based on” with “with.”

Done

Line 23: replace “status” with “growth stage”; as far as I can tell relative maturity isn’t a status. Replace “exploration” with “comparison.”

Done

Line 62: Replace “taxonomical” with “taxonomic.”

Done

Line 65: delete “purported”, unless you are doubtful of its uniqueness.

Done

Line 95: replace “like” with “such as.”

Done

Line 104: replace “examined” with “studied first hand” and modify the sentence accordingly.

Edit done and sentence rephrased to: “The subclades of theropods included in this dataset, apart from specimen studied first hand (DMNH 21183) are:”

Line 106: replace “performed” with “did.”

Done

Line 107: insert a comma after “measurements”; find an accurate replacement for “distilling”.

Replaced with ‘converting”

Line 108: replace “representing” with “that represents”; delete “to be.”

Done

Line 110: replace “estimate…of” with “compare”; replace “within” with “with.”

Done.

Line 208: is it “foramen” or “foramina”; the figure labels only one foramen. Please clarify.

Thanks for spotting this, it is foramina: clarified and edited in the relevant figure caption (Fig. 3)

Line 211: is there any evidence for the first alveolus? How do you justify the identification of the first preserved tooth as the second tooth in sequence? Clarify.

We reported in the description this explanation, following direct comparison with the condition in both Acheroraptor (Evans and Larson, 2013) and Saurornitholestes (Currie and Evans, 2020): “Given the position of the most erupted tooth (d3; Fig 3¬¬¬¬) above the Meckelian foramina (on the medial surface of the dentary, Fig 3B), and above an anteroventral process of the dentary (ave; Figs 3A and 3B), we identify this as the 3rd tooth in the dentary [e.g. 60, 61], with the anteriorly positioned, less erupted tooth (d2) being identified as the 2nd.”

Refs: 

Currie PJ, Evans DC. Cranial Anatomy of New Specimens of Saurornitholestes langstoni (Dinosauria, Theropoda, Dromaeosauridae) from the Dinosaur Park Formation (Campanian) of Alberta. Anat Rec. 2020; doi:10.1002/ar.24241

Evans DC, Larson DW, Currie PJ. 2013. A new dromaeosaurid (Dinosauria: Theropoda) with Asian affinities from the latest Cretaceous of North America. Naturwissenschaften. 100 (11): 1041-1049. doi: 10.1007/s00114-013-1107-5

Line 215: replace “shows” with “has.”

Done

Line 216: delete “of…tooth.”

Done.

Line 217: replace “dividing” with “separating.”

Done

Line 222: rephrase, “slightly anteroposteriorly longer than mediolaterally wide.”

Done

Line 224: replace “mid-way depth” with “midheight.”

Done

Line 225: replace “a” with “an.”

Done

Line 229: replace “, but” with “that is”; the word “but” is used to introduce an exception or contradiction, not a similarity or an elaboration upon a point.

Fixed accordingly

Line 231: replace “mesodistal” with “mesiodistal.”

Done

Line 233: replace “damaging” with “damage.”

Done

Line 235: insert “13” ahead of “denticles.”

Done

Line 236: delete “, with 13 denticles.”

Done

Line 239: word choice – does “coarser” mean “larger”? If so, say so to remove the ambiguity.

Changed accordingly with larger. 

Line 242: replace “observed” with “seen.”

Done

Line 244: replace “a hypothetical” with “an”; how was this estimate arrived at? Briefly justify.

Fixed and added: accounting for the half of the carina being inset in the tooth socket: for clarity.

Line 246: delete “morphological.”

Done

Line 253: what, exactly, is rounded and not pointed? I suspect this is a grammatical issue; please fix.

Replaced “rounded” with “with a sharp tip”

Line 263: replace “grains” with “grain.”

Done

Line 266. Replace ”in” with “along with.”

Done

Line 267: replace “points…21183” with “is evidence that DMNH 21183 is a juvenile.” Avoid passive voice – in science, it is ok to be declarative! In contrast, passive voice gives the impression that you are backing away from your evidence, which defeats the purpose of your hard work.

Thanks for pointing that out, changed accordingly.

Line 274: delete “traced and.”

Done.

Line 294: delete “combination.”

Done

Lines 294,295: “oval” and “lenticular” mean the same thing; pick one to avoid redundancy.

Replaced by just lenticular

Lines 300,301: replace “relatively better” with “well.”

Done.

Line 310: replace “between” with “among.”

Done

Lines 315,316: how is it that you can comment on the form of the first dentary tooth after you’ve made it clear that it is completely missing? Clarify.

Sentence removed.

Line 324: replace “observed” with “reported.”

Done.

Line 328: replace “with” with “have”; replace “compared” with “in contrast.” Also, does “coarser” just mean “larger?” Clarify.

All changed accordingly and larger chosen over “coarser”.

Line 333: replace “this” with “the”; delete “taxon.”

Done

Line 335: what about the ridge?!

Please see comment above.

Line 339: make sure that “anchiornithines” is the correct moniker to use. I.e., is there an “Anchiornithinae”?

Replaced with anchiornithid as Anchiornithidae has been favored by Hu et al. 2018 over the original definition of “Anchiornithinae” by Xu et al. 2016

Refs:

Xu; et al. (2016). "An Updated Review of the Middle-Late Jurassic Yanliao Biota: Chronology, Taphonomy, Paleontology and Paleoecology". Acta Geologica Sinica. 90 (6): 2229–2243. doi:10.1111/1755-6724.13033

Dongyu Hu; Julia A. Clarke; Chad M. Eliason; Rui Qiu; Quanguo Li; Matthew D. Shawkey; Cuilin Zhao; Liliana D’Alba; Jinkai Jiang; Xing Xu (2018). "A bony-crested Jurassic dinosaur with evidence of iridescent plumage highlights complexity in early paravian evolution". Nature Communications. 9 (1): Article number 217.

Line 343: replace “It….that” with “However,”; replace “been shown” with “occurred.”

Done

Line 345: pluralize “anchiornithid.”

Done

Lines 245 to 346: delete “, in….Eudromaeosauria.”

Done

Line 349: replace “reduction in” with “small” since there is no evolutionary or developmental process described here – you really don’t know if the size is plesiomorphic or the juvenile condition; avoid process-based terms when describing static, context-free morphology.

Thanks for the tip, and changed accordingly.

Line 353: replace “recalls a similar” with “is similar to the.”

Done

Line 354: pluralize “Velociraptorinae.”

Done

Line 357: “in-serrated/absent” are equivalent terms – pick one to reduce the annoyance of your reader. Concision is always appreciated in heavy osteodental descriptions!

Changed with unserrated.

Line 358: replace “observed” with “also seen in”; again, avoid passive voice.

Changed.

Line 365: replace “exhibit” with “have.”

Done.

Line 387: replace “are” with “include.’

Changed.

Line 392: delete “of the dentary.” Avoid redundancy.

Done

Line 394: replace “therefore” with “this criterion was”; delete “as…feature.”

Done

Line 395: delete “versus Dromaeosaurinae.”

Done

Line 397: briefly define DSDI/

Added “the ratio between number of mesial and distal denticles:

Line 402: is there a citation for the saurornitholestine teeth? What is their value?

Added and referenced relevant figure with DSDI plot

Lines 410-411: move these up to line 402 so that your reader can see how the saurornitholestine values compare right away. By the way, is there a difference between Saurornitholestinae and Velociraptorinae? Is see that Saurornitholestes is a velociraptorine, so Velociraptor is not a saurornitholestine?

Moved and modified also accordingly to some “in text” requests from Reviewer 1

Line 414: what is the DSDI? Give the value!

Added “outliers with DSDI around 2-3” and referenced relevant figure.

Lines 419 to 422: methods are best written in past tense. Please fix.

Fixed

Line 424: replace “provided” with “given.”

Done

Lines 434 to 437: fix the commas.

Commas removed and replaced by bracketed explanation

Line 436: replace “provided” with” represented.”

Done

Line 438: replace “presented” with “given.”

Done

Lines 444 to 446: didn’t you earlier state that the lateral ridge allies the specimen with Buitreraptor? How do you weight the dental data against the osteological evidence? Justify, especially since you are dealing with a juvenile specimen.

As mentioned in the response above, this has now been changed after new observations and reanalysis consequent to phylogenetic scoring. 

Line 449: please stop using “status” for “growth stage.”

Fixed

Lines 480 to 482: Given that dromaeosaurids are part of a clade of flying theropods, why couldn’t have they just flown along their migratory routes? Explain.

While we believe it is un-parsimonious to attribute volant habits to members of Eudromaeosauria, we improved the context of this section of the Discussion, backing up our interpretation that it is unlikely that these theropods nested in southern latitudes, then migrated to northern latitudes in the summer. The occurrence of a young individual of such a small-size itself is evidence that this individual wouldn’t have come from so far away as a migration entails, as it would have been dimensionally incapable of covering such large distance (e.g. Fiorillo & Gangloff 2001). The individual would not have at attained the minimum physical maturity necessary for such journey. Similar reasoning has been historically presented for hadrosaurid hatchlings by Horner (1982). While some basal members of Dromaeosauridae might have exhibited some sort of rudimentary (i. e. non-active flight), that might have been at best something close to what has been postulated for members of Microraptorinae (e.g. Sinornithosaurus and Zhenyuanlong (Xu et al, 1999; Alexander et al. 2010, Dyke et al. 2013), no such arguments have been made for saurornitholestines. Further, the anatomy of most dromaeosaurids, especially larger species, argues for a flightless lifestyle (Turner et al. 2007). That eliminates flight as an energy efficient means for these small predatory animals to cover the vast distances required to migrate across large wide geographic areas. Lastly, in response to a comment by one of the reviewers about the possibility of the young being volant, we point out that it is unlikely that the young forms migrated and the adult forms did not as there is no modern analog for such a model. We briefly discuss that within modern birds, the adults teach their young the necessary tools for migration (e.g. Baker, 1980).

Ref:

Horner JR. Evidence of colonial nesting and 'site fidelity' among ornithischian dinosaurs. Nature. 1982; 297(5868):675-676.

Dyke G, de Kat R, Palmer C, van der Kindere J, Naish D, Ganapathisubramani, B. Aerodynamic performance of the feathered dinosaur Microraptor and the evolution of feathered flight. Nature Communications. 2013; 4 (2489). doi:10.1038/ncomms3489

Alexander, D. E., Gong, E., Martin, L. D., Burnham, D. A. & Falk, A. R. Model tests of gliding with different hindwing configurations in the four-winged dromaeosaurid Microraptor gui. Proceedings of the National Academy of Science. 2010; 107, 2972-2976.

Xu X, Wang X-L, Wu X-C. A dromaeosaurid dinosaur with filamentous integument from the Yixian Formation of China. Nature. 1999; 401:262–266.

Turner AH, Pol D, Clarke J, Erickson G, Norell M. A basal dromaeosaurid and size evolution preceding avian flight. Science. 2007; 317:1378–1381.

Line 488: replace “southerner” with “southern.”

Done

Lines 485 to 488: why is large size evidence for “success”? Justify. What is meant by “success”? Clarify.

We removed this section as we in the end agreed with the consideration that most of this reasoning was too speculative and outside the scope of the study.

Line 493: change “stronger constrained” to “greater constraints.” What sort of constraints? Clarify.

Same as above

Lines 494 to 495: convoluted; fix.

Fixed.

Line 498: “eggs” should be singular.

Fixed

Lines 494 to 505: perhaps it is because it is late in the day, but I find this section to be very difficult to follow. Please redraft with a clearer argument and straightforward sentence structure.

Thanks for the suggestion. We redrafted this section to simplify the reasoning and expunge the speculative content as recommended by the reviewer.

Line 506: avoid passive voice; fix.

Fixed.

Lines 510 to 512: this can be shortened significantly; fix.

Done

Lines 513 to 515: the mere presence of taxon does not imply migration any more than it doesn’t imply migration. Fix.

Done

Line 523: should there be an “r” in “cfr”? Isn’t it just “cf.”? Where’s the period? Fix.

Fixed

Line 525: replace “appear” with “are.”

Fixed.

Line 526: replace “but” with “and” since you aren’t marking a difference or exception.

Fixed

Line 544: a period should follow “al”; e.g., Osborn et al. (1905); replace “for” with “that.”

Done

Line 545: insert “is” after the taxon name.

Added

Lines 513 to 555: I find this section to be, uncomfortably, speculative, but I leave it up to the judgment of the author and co-authors whether or not it needs to be reined in. Perhaps more citations in this section would put any concerns to rest. It just strikes me as a bridge too far when all of it is merely based on the relative abundance of teeth and – as far as I can tell - inconclusive functional inferences of tooth morphology. Please reconsider this section in a more critical light.

See comments above.

Lines 560 to 562: You don’t really know that it wasn’t migratory; it could be that all small dromaeosaurids were fully capable of flight early in growth. Please reconsider this point more critically – does the inference really have support?

See comments above.

Lines 564 to 565: again, you have no rationale to think this – if dromaeosaurids were volant, there’s literally nothing to stop them from migrating. It’s possible flight brought deinonychosaurians their global distribution. Regardless, I don’t think you have the evidence to make the claim, unless there’s something obvious that I missed earlier.

See comments above.

Figure 3: “a3” is not labeled; label both interdental plates; label the circular fossa.

Done.

---

## [Decision Letter · Decision Letter 1]

3 Jun 2020

PONE-D-20-05495R1

The first juvenile dromaeosaurid (Dinosauria: Theropoda) from Arctic Alaska

PLOS ONE

Dear Dr. Chiarenza,

Thank you for submitting your manuscript to PLOS ONE. After examining a revised version of the manuscript, both reviewers were impressed by the changes made and agree that the manuscript is substantially improved; both reviewers also identified a few final minor points to be revised prior to acceptance. We invite you to submit a revised version of the manuscript that addresses these points raised during the second review process.

Please see the reviewers' comments below. Their final suggested changes are minor, although I would particularly encourage the authors to consider Reviewer 2's comments regarding specimen numbers in Dataset S2, if this is possible.

We look forward to receiving your revised manuscript.

Kind regards,

Laura Beatriz Porro, Ph.D.

Academic Editor

PLOS ONE

Reviewers' comments:

Reviewer's Responses to Questions

**Comments to the Author**

1. If the authors have adequately addressed your comments raised in a previous round of review and you feel that this manuscript is now acceptable for publication, you may indicate that here to bypass the “Comments to the Author” section, enter your conflict of interest statement in the “Confidential to Editor” section, and submit your "Accept" recommendation.

Reviewer #1: All comments have been addressed

Reviewer #2: All comments have been addressed

2. Is the manuscript technically sound, and do the data support the conclusions?

Reviewer #1: Yes

Reviewer #2: Yes

3. Has the statistical analysis been performed appropriately and rigorously? 

Reviewer #1: Yes

Reviewer #2: I Don't Know

4. Have the authors made all data underlying the findings in their manuscript fully available?

Reviewer #1: Yes

Reviewer #2: Yes

5. Is the manuscript presented in an intelligible fashion and written in standard English?

Reviewer #1: Yes

Reviewer #2: Yes

6. Review Comments to the Author

Reviewer #1: Chiarenza and colleagues have considerably improved their MS and, from what I can see, they have addressed pretty much all suggestions provided by both reviewers. I congratulate them for that and thank them for taking all our remarks into consideration. I also praise them for using the most recent techniques to identify their material, for being so thorough in the discussion regarding the implication of the presence of saurornitholestines in high latitudes, and for using an exhaustive literature. I only provided minor suggestions and corrections in the pdf version of their paper, which is definitely ready to go according to me. There are only three things I wish to see in this paper that appear to be missing:

1) Please provide some information on the ecosystem surrounding this juvenile saurornitholestine. What other dinosaurs were living at that time and at such latitude? What about the flora? I may have missed that from the text but it would be great to briefly discuss what could be the potential prey of this dromaeosaurid.

2) Please provide the paleogeographic coordinates of the specimen (using the PBDB) to illustrate the high latitude it was living in.

3) You finally propose a perennial residency for saurornitholestine dromaeosaurids in high latitude, the most important result of this study. Was it already proposed for theropods and dromaeosaurids by other authors? If yes, please cite the relevant literature.

Kind regards,

Christophe Hendrickx, San Miguel de Tucuman, the first of June 2020.

Reviewer #2: I congratulate the authors on bringing rigorous methods to bear on the identification of the specimen. The dinosaur-bearing deposits of the Prince Creek Formation (PCF) are important in that they document the stable, quiescent conditions that preceded the disruptive dispersal of Tyrannosaurus rex into Laramidia from Asia.

I am satisfied with the changes that the authors have made, which have resulted in a robust hypothesis against which future fossil discoveries in the PCF will be compared.

I am satisfied with the results of the cladistic analyses, since I am a user of TnT; however, I have not used PCA analyses and so I cannot pass judgement on that part of the work.

Caveat emptor: Since we are under COVID-19 conditions, I have had to review the revisions on screen: a few weeks ago my printer ran out of toner and, resulting from delivery snafus, I am still awaiting a re-ordered cartridge. The point is that I have reviewed the article on screen, which raises the probability that I have missed minor typos and other small errors.

Sincerely,

Thomas D. Carr, PhD

Associate Professor of Biology

Carthage College

Kenosha, WI

Specific comments

101: “crow” should be “crown”.

238: “emplaced” is not needed, please delete.

288: “mesodistal” should be “mesiodistal”.

411 to 443: this para is exhaustingly long to read; please cut it into two.

435: delete “more” since it is used again in the same sentence; overall, try to avoid the word, since it brings in passive voice.

470: please use a more precise word than “coarser”; do you mean a high number of denticles or a low number of denticles?

567: replace “erected” with “first named” or “first coined”. Names are not the result of erections.

References cited: the indentation format is inconsistent and changes at reference #100.

S1 Dataset caption: changes “Systematics” to “Sytematic”.

Dataset S2: there are no specimen numbers associated with the measurements; I leave it to the editor to decide if the numbers are required, in the interests of reproducibility. Perhaps these data are from another source and the specimen numbers are there; if so, then the authors should explicitly mention that somewhere that is easy to find in the ms.

7. PLOS authors have the option to publish the peer review history of their article (what does this mean?). If published, this will include your full peer review and any attached files.

Reviewer #1: Yes: Christophe Hendrickx

Reviewer #2: Yes: Thomas D. Carr

---

## [Author Response · Author response to Decision Letter 1]

4 Jun 2020

Dear Editor, 

Thanks for editing our manuscript entitled “The first juvenile dromaeosaurid (Dinosauria: Theropoda) from Arctic Alaska”. Attached to this letter we provide a revised version of the Manuscript with updated figures and Supplementary material. We accommodated all the requests from the reviewers. Other than a point by point response below, we have included all suggestions from Christophe Hendrickx on some minor typos which we had previously missed. Figure 3 has been updated to conform to the nomenclature suggested by the Reviewer on the denomination of the positions of the teeth (rdt). Captions have been updated accordingly. Additional information for Dataset S2 have been included in the relative caption following Thomas Carr’s recommendation. We would like to thank both reviewers for their thorough contribution which greatly improved this study from its original version. An additional comment to the Editor: during the last round of review, both the first (Alessandro Chiarenza) and the second author (Tony Fiorillo) changed affiliation. We reported these changes in the new version of the manuscript.

Below is outlined a point by point response to both Reviewers. Whereas their text has been highlighted in italics, our response has been written in bold.

Best Regards,

Alfio Alessandro Chiarenza

On behalf of all coauthors.

Reviewer #1: Chiarenza and colleagues have considerably improved their MS and, from what I can see, they have addressed pretty much all suggestions provided by both reviewers. I congratulate them for that and thank them for taking all our remarks into consideration. I also praise them for using the most recent techniques to identify their material, for being so thorough in the discussion regarding the implication of the presence of saurornitholestines in high latitudes, and for using an exhaustive literature. I only provided minor suggestions and corrections in the pdf version of their paper, which is definitely ready to go according to me. There are only three things I wish to see in this paper that appear to be missing:

1) Please provide some information on the ecosystem surrounding this juvenile saurornitholestine. What other dinosaurs were living at that time and at such latitude? What about the flora? I may have missed that from the text but it would be great to briefly discuss what could be the potential prey of this dromaeosaurid.

Thanks for the suggestion. We added the passage below before the final paragraph of the paper. ‘This Alaskan Saurornitholestinae would have lived in a biotope featuring a coniferous open woodland (dominated by taxodiaceous conifers) with an angiosperm-fern understory [61, 160, 161]. Herbaceous vegetation included ferns, angiosperms, abundant horsetails and other sphenophytes [61, 160, 161]. This ancient Arctic ecosystem would have included animals such as basal ornithopods [21], the hadrosaurid Edmontosaurus [162], the centrosaurine Pachyrhinosaurus [22], the diminuitive tyrannosaurid Nanuqsaurus [23], a large troodontid [26] and at least another dromaeosaurid taxon closer to Dromaeosaurus [28] than to Saurornitholestes. Small body-sized animals representing potential preys for the Arctic saurornitholestine (Fig 10) might have been mammals such as the methatherian Unnuakomys [67], a Gypsonictopidae and the multituberculate Cimolodon [25, 163].’

2) Please provide the paleogeographic coordinates of the specimen (using the PBDB) to illustrate the high latitude it was living in.

Paleocoordinates for the specimen added to figure 1 after modern-day coordinates as “(Paleocoordinates from paleobiodb.org: N 89.13°, W -104.73°)”

3) You finally propose a perennial residency for saurornitholestine dromaeosaurids in high latitude, the most important result of this study. Was it already proposed for theropods and dromaeosaurids by other authors? If yes, please cite the relevant literature.

References (20, 25, 26, 28) added in the sentence of the discussion “Taken together, we infer that DMNH 21183 implies a perennial residency of this dromaeosaur clade (Saurornitholestinae) in the Arctic [20, 25, 26, 28]”.

Kind regards,

Christophe Hendrickx, San Miguel de Tucuman, the first of June 2020.

Reviewer #2: I congratulate the authors on bringing rigorous methods to bear on the identification of the specimen. The dinosaur-bearing deposits of the Prince Creek Formation (PCF) are important in that they document the stable, quiescent conditions that preceded the disruptive dispersal of Tyrannosaurus rex into Laramidia from Asia.

I am satisfied with the changes that the authors have made, which have resulted in a robust hypothesis against which future fossil discoveries in the PCF will be compared.

I am satisfied with the results of the cladistic analyses, since I am a user of TnT; however, I have not used PCA analyses and so I cannot pass judgement on that part of the work.

Caveat emptor: Since we are under COVID-19 conditions, I have had to review the revisions on screen: a few weeks ago my printer ran out of toner and, resulting from delivery snafus, I am still awaiting a re-ordered cartridge. The point is that I have reviewed the article on screen, which raises the probability that I have missed minor typos and other small errors.

Sincerely,

Thomas D. Carr, PhD

Associate Professor of Biology

Carthage College

Kenosha, WI

Specific comments

101: “crow” should be “crown”.

Fixed.

238: “emplaced” is not needed, please delete.

Removed accordingly.

288: “mesodistal” should be “mesiodistal”.

Fixed accordingly.

411 to 443: this para is exhaustingly long to read; please cut it into two.

Done at the beginning of the sentence starting with ‘The lenticular shape of…’

435: delete “more” since it is used again in the same sentence; overall, try to avoid the word, since it brings in passive voice.

Thanks for the suggestion, ‘more’ removed accordingly.

470: please use a more precise word than “coarser”; do you mean a high number of denticles or a low number of denticles?

‘Coarser’ replaced with ‘less’ for clarity

567: replace “erected” with “first named” or “first coined”. Names are not the result of erections.

Changed accordingly.

References cited: the indentation format is inconsistent and changes at reference #100.

We are aware of this issue, but this is unfortunately an automatic setting of Microsoft Words which the user seems to be unable to change. Editorial teams are probably able to remove this issue formatting indentation accordingly to the journal’s needs at the production stage. 

S1 Dataset caption: changes “Systematics” to “Sytematic”.

Fixed.

Dataset S2: there are no specimen numbers associated with the measurements; I leave it to the editor to decide if the numbers are required, in the interests of reproducibility. Perhaps these data are from another source and the specimen numbers are there; if so, then the authors should explicitly mention that somewhere that is easy to find in the ms.

The reviewer is correct in pointing out that the datasets are primarily available and accessible in the cited papers. We included to the Dataset S2 caption the sentences: ‘Systematic entries follow methodology as described in the Material and Methods section. Original datasets with specimen-level denominations can be found in Gerke and Wings (https://doi.org/10.1371/journal.pone.0158334.s001) and Larson and Currie (https://doi.org/10.1371/journal.pone.0054329.s001).’

---

## [Editor Report · Decision Letter 2]

9 Jun 2020

The first juvenile dromaeosaurid (Dinosauria: Theropoda) from Arctic Alaska

PONE-D-20-05495R2

Dear Dr. Chiarenza,

Thank you very much for taking all of the reviewers comments into consideration and carefully revising this manuscript. We’re pleased to inform you that your manuscript has been judged scientifically suitable for publication and will be formally accepted for publication once it meets all outstanding technical requirements. 

Kind regards,

Laura Beatriz Porro, Ph.D.

Academic Editor

PLOS ONE
---

## [Editor Report · Acceptance letter]

11 Jun 2020

PONE-D-20-05495R2 

The first juvenile dromaeosaurid (Dinosauria: Theropoda) from Arctic Alaska 

Dear Dr. Chiarenza:

I'm pleased to inform you that your manuscript has been deemed suitable for publication in PLOS ONE. Congratulations! Your manuscript is now with our production department. 

Kind regards, 

on behalf of

Dr. Laura Beatriz Porro 

Academic Editor

PLOS ONE